# Split luciferase-based assay to detect botulinum neurotoxins using hiPSC-derived motor neurons

Laurent Cotter[1], Feifan Yu[2,3,4], Sylvain Roqueviere[1], Juliette Duchesne de Lamotte[1], Johannes Krupp[5], Min Dong [2,3✉] & Camille Nicoleau [1✉]

Botulinum neurotoxins (BoNTs) have been widely used clinically as a muscle relaxant. These toxins target motor neurons and cleave proteins essential for neurotransmitter release like Synaptosomal-associated protein of 25 kDa (SNAP-25). In vitro assays for BoNT testing using rodent cells or immortalized cell lines showed limitations in accuracy and physiological relevance. Here, we report a cell-based assay for detecting SNAP-25-cleaving BoNTs by combining human induced Pluripotent Stem Cells (hiPSC)-derived motor neurons and a luminescent detection system based on split NanoLuc luciferase. This assay is convenient, rapid, free-of-specialized antibodies, with a detection sensitivity of femtomolar concentrations of toxin, and can be used to study the different steps of BoNT intoxication.

[1] IPSEN Innovation, 5 avenue du Canada, 91940 Les Ulis, France. [2] Department of Urology, Boston Children's Hospital, Boston, MA 02115, USA. [3] Department of Microbiology, Harvard Medical School, Boston, MA 02115, USA. [4] AlphaThera, Philadelphia, PA 19146, USA. [5] Recherche Servier, 125 chemin de ronde, 78290 Croissy-sur-seine, France. ✉email: min.dong@childrens.harvard.edu; camille.nicoleau@ipsen.com

Botulinum neurotoxins (BoNTs) are a family of bacterial toxins[1–3]. Seven major serotypes (named BoNT/A to G) can be distinguished based on their immunological differences and further divided into subtypes based on their amino acid sequence differences[4]. BoNTs target peripheral motor neurons and block the release of acetylcholine by cleaving members of the soluble N-ethylmaleimide-sensitive fusion protein attachment protein receptor (SNARE) complex, which is essential for neurotransmitter release. This leads to muscular relaxation and flaccid paralysis[5]. BoNTs are used in the clinic to treat muscle dysfunction and an increasing number of other indications[6, 7]. The toxin is produced as a single polypeptide chain of 150 kDa which is post-translationally cleaved. This yields a di-chain consisting of a 100 kDa heavy chain (HC) and a 50 kDa light chain (LC) linked by a disulfide bond.

The intoxication mechanism involves at least four steps: receptor binding by the HC, HC and LC internalization in endosomes, LC translocation into the cytosol and finally specifically cleavage of neuronal SNARE proteins by the LC[2]. More precisely, the HC binds selectively to receptors at the presynaptic surface. The binding is initiated by the recognition of complex gangliosides such as GT1b and GD1a[6, 8, 9] followed by the binding to a neuronal protein receptor such as synaptic vesicle proteins SV2 or Synaptotagmin I/II[2, 10–20]. This complex is then internalized into the cell by endocytosis. After acidification of the endosomal vesicle, the LC is translocated into the cytoplasm and it acts as an endopeptidase to cleave its substrate SNARE proteins.

Like all pharmaceutical products, BoNTs require accurate and reliable measurement of their activity. For years, the gold standard assay to test BoNT pharmaceuticals has been the mouse lethality bioassay (MLB)[21]. In this in vivo assay, increasing doses are injected in animals and the dose, which is responsible for the death of half of the population, called median lethal dose ($LD_{50}$) is determined. This method utilizes a large number of animals and does not follow the principle of the 3Rs (reduction, replacement and refinement) of animal use in research. Therefore, the development of alternative animal-free methods is needed.

Several in vitro cell-based assays (CB-assay) have been developed for BoNT potency assays to avoid the use of animals for this purpose[22–26]. The main advantages of these CB-assays over LD50 determination are more quantitative readout and a better reproducibility.

The first replacement assay approved by the FDA for analyzing BoNT/A was developed by Fernandez et al. and relies on a specific monoclonal antibody that recognizes only the cleaved form of SNAP-25, a member of neuronal SNARE proteins and a substrate for BoNT/A, C, and E. It was developed in the human SiMa cell line with an ELISA-based format[22]. Other approaches have been developed as well such as utilizing two fluorescent proteins (CFP and YFP) linked with a SNARE protein fragment[23]. Finding suitable cells is a crucial point to develop this type of CB-assay. Human-induced pluripotent stem cells (hiPSC)-derived neurons are a good cellular model. Indeed, several studies have shown that these cells are a highly sensitive model to test BoNTs, especially hiPSC-derived motor neurons (MNs) that represent the most physiological system to study BoNTs[27–31].

In this work, we describe the development of a CB-assay combining human iPSC-derived MNs with an antibody-free detection system using the well-known NanoLuc[32] binary Technology (NanoBiT) to detect toxin action. NanobiT[33] is a luciferase-based complementation reporter system used for protein–protein interaction and several CB-assay have been developed using this technology[34–36] to undercover cellular mechanism or for drug discovery[37].

For this study, we developed a toxin sensor constituting two complementary fragments of NanoLuc luciferase linked through fragments of SNARE proteins. In the presence of toxins, the linker region is cleaved, and the two luciferase pieces are separated resulting in a decrease in luminescence emission.

The assay presented here offers the possibility to test toxins after an acceptable timeframe of cell culture (9 days) in physiologically relevant cells and can be read directly after toxin treatment. Its sensitivity allows us to detect, compare, and rank the potency of different BoNTs in the femtomolar range.

## Results

**Development of an assay based on a split form of NanoLuc luciferase to detect BoNT activity.** In order to establish a cell-based approach to detect the activity of BoNT, we first developed an antibody-free detection system using recently described split NanoLuc luciferase[33]. More precisely, we expressed a single polypeptide chain composed of NanoLuc luciferase split into two parts and maintained close to each other by a linker containing fragments of neuronal SNARE proteins. Once BoNT cleaves its target, the two parts of the NanoLuc luciferase are no longer in proximity, resulting in a decrease in its activity (Fig. 1a).

To test if the remaining luciferase activity could be inversely correlated with the toxin activity, we first designed acellular sensor protein probes. Two single-chain polypeptides containing different BoNT substrates were designed (Fig. 1b). One sensor contains a linker composed of a SNAP-25 fragment (residues 141-206) that can be cleaved by BoNT/A, E and C; and the second version uses a VAMP1 fragment (residues 35–96) as the linker, which can be cleaved by BoNT/B, D, F and G. These sensors proteins were purified as recombinant proteins.

Then, to evaluate the functionality of these sensor proteins and their respective sensitivity to BoNT, the two versions of sensors were incubated with a gradient of concentration of BoNT/A or BoNT/B for 24 h in test tubes and the remaining luciferase activity was measured.

The sensor containing a SNAP-25 fragment showed a concentration-dependent decrease in luciferase activity after BoNT/A incubation (Fig. 1c). BoNT/B did not affect the luciferase activity of this sensor, demonstrating its specificity for BoNT/A (Supplementary Fig. 1).

The second sensor containing VAMP1 fragment showed a concentration-dependent decrease of luciferase activity after BoNT/B incubation (Fig. 1d). Incubation of this sensor with BoNT/A did not affect the luciferase activity, demonstrating its specificity for BoNT/B (Supplementary Fig. 1).

**Detect BoNT/A in cells and human motor neurons.** We next sought to develop cell-based assays using split NanoLuc luciferase-based sensors. As the majority of BoNT products used in clinics are BoNT/A, we focused on establishing a SNAP-25-cleaving cell-based sensor. For this purpose, a construct expressing a split NanoLuc luciferase maintained together with a linker containing the full length of SNAP-25 was developed. In addition to this SNARE, the full-length sequence of the firefly luciferase was added on the N-terminal side of SNAP-25. The addition of the firefly luciferase provides an internal control for the measurement of the level of expression of the polypeptide. This recombinant reporter was named BoNT sensor4 and is illustrated in Fig. 2a. Then, this sensor protein was expressed in neurons via lentiviral particles, as developed by Gascon et al.[38] and adapted by us for co-expressing BoNT sensor4 and GFP. The lentiviral expression vector is illustrated in Supplementary Fig. 2a.

Primary rat cortical neurons transduced with these lentiviral particles were exposed to various concentrations of BoNT/A and cell lysates were harvested 48 h later. The luciferase activities of both firefly and NanoLuc™ luciferase were measured. Incubating neurons

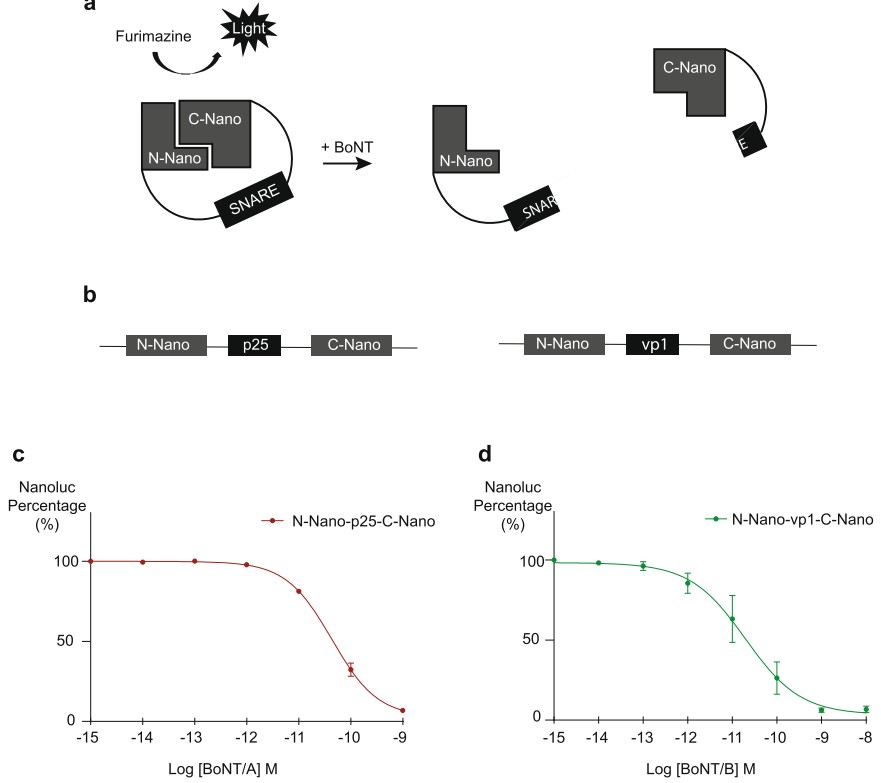

**Fig. 1 Split luciferase sensors for detecting BoNT activity. a** Schematic representation of the split luciferase-based biosensor. The split NanoLuc™ luciferase is linked together by a linker containing a SNARE sequence. By cleaving the SNARE part of the sensor, BoNTs separate NanoLuc™ luciferase and prevent it from reconstitution into a functional luciferase. The decrease in luminescence is inversely proportional to the toxin activity. **b** Schematic drawing of two sensor proteins designed for BoNTs detection. For SNAP-25 (p25)-cleaving BoNT, the sensor N-Nano-p25-C-Nano was prepared. For VAMP1(vp1)-cleaving BoNT, the sensor protein N-Nano-vp1-C-Nano was prepared. **c** Acellular detection assay of BoNT/A. Sensor proteins (30 nM) were mixed with different concentrations of BoNT/A. The curve was plotted based on the percentage of luminescence of NanoLuc luciferase by using toxin added sample/control sample. Assay was performed in duplicate. **d** Acellular detection assay of BoNT/B. The process was the same as described for detecting BoNT/A. Assay was performed in duplicate.

with toxin reduced the signal of NanoLuc™ luciferase, showing BoNT sensor4 ability to sense toxin activity in cells (Fig. 2b).

We next decided to transfer this assay into a well characterized, commercially available hiPSC-derived model, the iCell Motor Neurons (MNs). This human neuronal model has been shown to be highly sensitive to BoNT/A and BoNT/E, and to express, after 14 days of culture, markers of authentic and functional motor neurons[30]. First, we checked the biosensor expression 14 days after thawing, using lentiviral particles. We transduced iCell MNs and after 2 weeks of culture, we observed that all MNs were GFP positive (Supplementary Fig. 2b), indicating that lentiviral particles are efficient in transducing these human cells. After cell lysis we checked the expression of the engineered polypeptide by western blot using an antibody against firefly luciferase. This blot revealed a single band at the expected molecular weight suggesting that the SNAP-25 sensor is well expressed (Supplementary Fig. 2c).

Because qPCR analysis performed previously in our group indicated that iCell MNs expressed a high level of the MNs markers, Choline Acetyl Transferase and ISLET1, and proteins involved in BoNT intoxication one week after thawing[30], we compared toxin sensitivity after 1 or 2 weeks of culture. At these two different stages of maturation, we treated cells with BoNT/A for 24 h and we tested SNAP-25 cleavage by western blot. At the two timepoints tested, we observed a concentration-dependent increase in SNAP-25 cleavage with a complete cleavage of the target at higher concentrations. The two curves were similar with a comparable pEC50 (11.89 ± 0.03 at day 7 and 11.89 ± 0.04 at day 14–15; $EC_{50} = 1.35$; pM = 3.6 U/well) indicating a similar sensitivity to the toxin (Supplementary Fig. 3

and Fig. 2c). This result indicates that the extra week in culture did not change toxin sensitivity of MNs.

To be able to use the iCell MN as soon as 1 week after thawing for our assay, we finally checked, at an early maturation timepoint, the impact of transduction on the cell culture. After 8 days of culture, we fixed the cells and quantified the number of cells, the transduction rate by looking at the expression of firefly luciferase and the number of MNs using the nuclear motoneuronal marker ISLET1. As expected, the use of a lentivirus induced some cell death in our culture. A decrease of 40% on cell population was observed in transduced cells (Supplementary Fig. 2d) with a transduction rate of 66% (firefly+ cells/number of cells). More importantly, this analysis showed that, at this stage, the fraction of ISLET1+ cells expressing firefly luciferase (ISLET1+ and Firefly+ cells) is similar to the fraction of ISLET1+ cells in non-transduced cultures (Fig. 2d, e). Although transduction has an impact on cell survival, no negative impact on the motoneuronal identity of the cell culture was detected, showing that transduction does not impair motor neurons differentiation.

Altogether these results confirm that iCell MNs overexpress BoNT sensor4 and already after 7 days in culture can be used for a cell-based assay without losing toxin sensitivity.

**Femtomolar concentrations of BoNT/A or /E can be measured in human MNs expressing BoNT sensor4.** Based on these results we established a CB-assay protocol that can be run in 9 days, as summarized in Fig. 3a. Briefly, iCell MNs are transduced 48 h

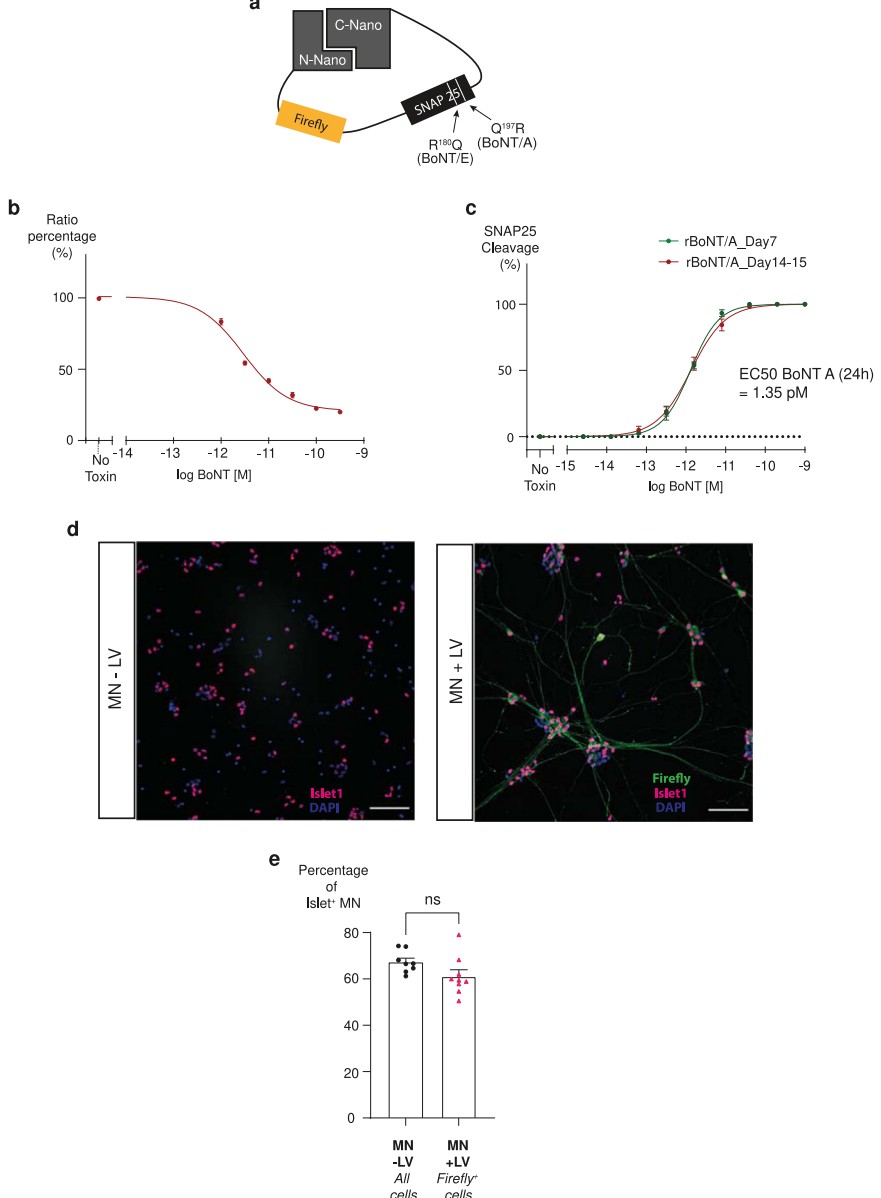

**Fig. 2 Split NanoLuc™ luciferase sensor can be used in rodent and human neurons. a** Schematic representation of BoNT sensor4 construct. The split NanoLuc™ luciferase is linked together by a linker containing the Firefly luciferase and SNAP-25. The cleavage sites for BoNT/A and E are indicated. **b** Detection assay of BoNT/A using virus-transduced rodent neuronal cells. The remaining luciferase activity was measured. Assay was performed in triplicate. **c** Comparison of iCell MN toxin sensitivity to BoNT/A at day 7 (green circles) and day 14–15 (red circles). Graph shows the dose–response curve of SNAP-25 cleavage. Data points are mean of $n = 3$ independent experiments performed in triplicate. **d, e** Impact of lentiviral transduction (MOI 16) on MN identity 6 days after transfection. ISLET1-positive and Firefly-positive MNs were counted. **d** Immunostaining of DAPI (blue), Islet1 (purple) and Firefly (green) in iCell MN. Scale bar 100 μm. **e** Graph showing quantification of the percentage of Islet positive cells (Islet+ cells/DAPI+ cells) in the non-transduced conditions (black circles) and the percentage of Islet positive cells in the fraction of Firefly-positive cells in the transduced conditions (pink triangles). The percentage of Islet + cells in the indicated conditions represented a mean of 8 wells per conditions (≥750 cells were counted per well). Statistical two-tailed Student's t-tests: *$P < 0.05$; **$P < 0.01$; ***$P < 0.001$; NS, $P > 0.05$ not significant (NS). Error bars show SEM.

after plating, followed by 6 days of further cell culture. Then, a 24 h toxin treatment 8 days after first plating is done, after which luminescence signals can be directly measured.

To evaluate the sensitivity of this CB-assay, we established concentration–response curves for two different recombinant BoNT serotypes, BoNT/A and BoNT/E. Both toxins showed a concentration-dependent decrease in the luminescent signal. The remaining NanoLuc luciferase signal could be inversely correlated with the toxin activity (Fig. 3b). Maximal activity was observed at the highest concentration for both toxins. Western blots performed at this concentration showed that the

sensor is completely cleaved by BoNT/A and E (Supplementary Fig. 4a).

As shown by us previously using a western blot SNAP-25 cleavage assay[30], BoNT/E ($pEC_{50} = 13.45 \pm 0.03$; $EC_{50} = 0.036$; pM = 0.1 U/well) was significantly more potent than BoNT/A ($pEC_{50} = 12.55 \pm 0.13$; $EC_{50} = 0.28$; pM = 0.79 U/well) in this assay (Fig. 3c). As a control, a mutant form of BoNT/A with inactivated LC did not lead to any changes in luminescence.

**Assessing binding and translocation of BoNT.** To determine if the CB-assay could also be used to study the mode of action of

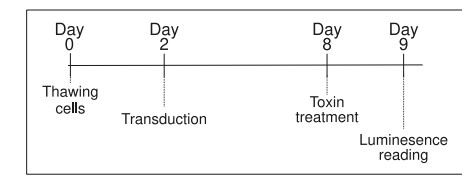

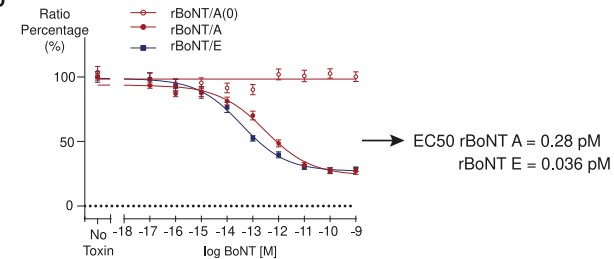

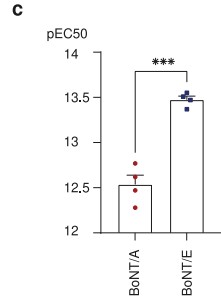

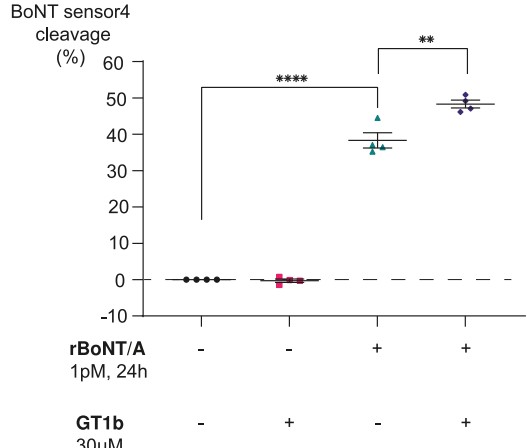

**Fig. 3 Cell-based assay using the split NanoLuc™ luciferase sensor.**
**a** Protocol of the CB-assay using the BoNT sensor4. **b**, **c** Comparison of
rBoNT/A (0) (red empty circles), rBoNT/A (red circles) and rBoNT/E
(blue squares) potency in the BoNT sensor4 CB-assay. Data points are
mean of $n = 4$ independent experiments in triplicate. **b** Graph shows
concentration–response curves. **c** Comparison of the pEC50 of rBoNT/A
and E calculated from curves above. Statistical two-tailed Student's $t$-tests:
$*P < 0.05$; $**P < 0.01$; $***P < 0.001$; NS, $P > 0.05$ not significant (NS). Error
bars show SEM.

BoNT, we evaluated the effect of two well-known modulators of
BoNT binding and LC translocation.

It was previously shown that the addition of ganglioside GT1b, a
BoNT coreceptor, could improve toxin sensitivity of hiPSC-derived
neurons[26] by facilitating BoNT binding. To evaluate if our CB-
assay would be sensitive to gangliosides, we added 30 µM of GT1b
to the cell culture, 48 h before treatment with BoNT/A. First, we
showed that the addition of GT1b to the media, effectively
increases the amount of this ganglioside at the surface of neurons
(Supplementary Fig. 6) and this addition slightly improves BoNT/
A efficiency to cleave the engineered target as shown in Fig. 4a.

Moreover, using ConcanamycinA (ConA), a specific vacuolar
$H^+$-ATPase inhibitor, which blocks endosomal acidification and
therefore antagonizes the action of BoNT[39], we tested if our assay
is sensitive to this effect. Figure 4b shows that treating transduced
MNs with 1 nM BoNT/A in high-$K^+$ concentration conditions
(to stimulate synaptic vesicle exocytosis), for 10 min, resulted in
approximately 30% substrate cleavage during a subsequent 4 h.
This effect was completely blocked when 250 nM of ConA was
co-administrated with the toxin to cells.

Together, our data show that the cell-based assay described
here can be utilized to characterize some intoxication steps in
physiologically relevant human MN models.

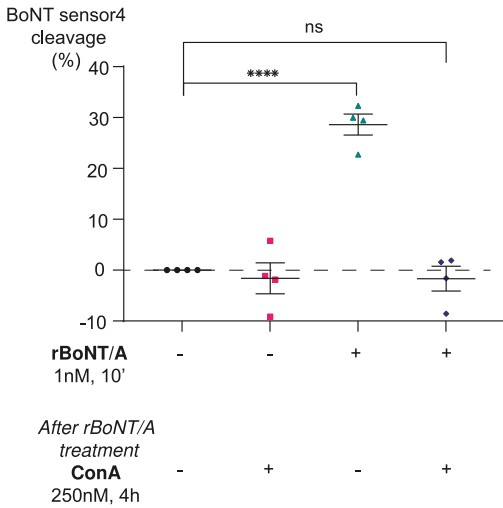

**Fig. 4 Mechanistic assay using the CB-assay. a** Increasing cell uptake by
GT1b addition. rBoNT/A (1 pM) was incubated 24 h in the absence (green
triangles) or presence of GT1b (blue diamonds) and the action of the toxin
was determined by measuring the percentage of BoNT sensor4 cleavage.
Data points are plotted as individual values for each independent
experiment and mean (4 experiments performed in triplicates). Controls
are cells with no treatment corresponding to basal conditions (black circles)
and cells incubate only with GT1b (pink squares). **b** Blocking of toxin action
by ConA. rBoNT/A (1 nM) was incubated 10 min in stimulating conditions
(60 nM KCl). After toxin washout and 4 h in the absence (green triangles)
or presence (blue diamonds) of ConA, the percentage of BoNT sensor4
cleavage was determined. Data points are plotted as individual values for
each independent experiment and mean (4 experiments performed in
triplicates). Controls are cells with no treatment corresponding to basal
conditions (black circles) and cells incubate only with ConA (pink squares).
Data points are mean of $n = 4$ independent experiments in triplicate.
Statistical ordinary one-way ANOVA multiple comparisons: $*P < 0.05$;
$**P < 0.01$; $***P < 0.001$; $****P < 0.0001$; NS, $P > 0.05$ not significant (NS).
Error bars show SEM.

## Discussion

Here we describe a hiPSC cell-based assay that provides a reproducible and highly sensitive model for SNAP-25 cleaving toxins and which is independent of using any specialized antibodies. The assay combines human iPSC-derived MNs, which represent a physiological cellular model for BoNT activity[27–31], with a straightforward, simple detection of the enzymatic action of the toxin activity by directly quantifying luminescence after toxin treatment.

Several studies already demonstrated the potential of hiPSC-derived neurons and MNs in particular for BoNT testing[27–31]. More precisely, it has been shown that MNs are a sensitive neuron type for BoNTs[28–30].

The manufacturer that provides the cells advised to culture the cells for 14 days in order to obtain mature MNs. In the literature, previous studies suggested that there is a time window for BoNT testing after hiPSC-derived neurons thawing. Withemarsh et al.[27] analyzed BoNT sensitivity by SNAP-25 cleavage and they showed no differences in BoNT sensitivity of neurons cultured for 4–14 days. In our group, previous results[30] indicated that the MN-derived neurons used in this study express marker of functional motor neurons after 14 days of culture but these cells express a high level of MN markers and protein involved in BoNT intoxication 1 week after thawing.

If 14 days are required to obtain highly mature and functional MNs, we confirmed here that we could test the toxin from day 7 without losing sensitivity.

The "direct" luminescent readout used here enabled measurement of BoNT activity less than one hour after toxin treatment. Compared to a cell-based assay with an ELISA readout[22], the assay described here does not need several steps of incubation and wash due to the use of antibodies. Furthermore, it does not require any special antibody which can be time consuming and difficult to produce. Overall, our assay is easier and faster than an ELISA-based readout.

Alternative assays have been developed with fluorescent probes in an effort to avoid using antibodies[23], but these probes need to be stimulated with light. This stimulation tends to produce a very bright signal with a high background. In contrast, the readout used here (chemiluminescence) produces a less bright signal with less background[40]. O'Brian and all showed that chemiluminescent assay done in multiwell plates provides better results compared to analogous fluorescent assays[41]. Alternative readouts based on natural luminescence[24] or on electrochemical multi-electrode[25] arrays have been developed. Pathe-Neuschafer-Rube et al. developed an assay based on targeting Gaussia luciferase to secretory vesicles in the human SiMa cell line and showed that BoNT/A could inhibit luciferase secretion[24]. These approaches are able to detect all BoNT serotypes but they are still not sensitive enough to be used as a cell-based assay.

Even though we designed a specific SNAP-25-cleaving BoNT assay our system is very versatile and BoNT target sequence can be easily changed. The production of in vitro sensor targeting VAMP1 illustrates that point.

To be a good in vitro alternative to the mouse lethality bioassay (MLB), assay sensitivity is a crucial parameter. In our study, $EC_{50}$ of 0.28 pM (~0.79 U per well) was determined for BoNT/A. Two recent studies (Pellett et al., 2019[28] and Schenke et al., 2020[29]) using human iPSC-derived motor neurons have reported different BoNT potencies. The study published by Johnson's lab evaluated BoNT/A $EC_{50}$ at 0.006 U/well and Seeger's lab estimated it at 0.046 pM. These discrepancies in BoNT/A potencies determination amongst these studies may be due to several reasons. First, the natures and sources of the tested BoNTs were not the same. Both Pellett's and Schenke's studies used BoNT produced by Clostridium while BoNT tested in our study was recombinantly produced by *Escherichia coli*. Second, durations of exposure to toxins were not the same. Pellett's and Schenke's studies treated the cells for 48 h while we used a treatment of 24 h. Nevertheless, the sensitivity of our assay is equivalent to the one obtained by the first CB-assay approved by the FDA[22].

Moreover, the lowest dose that causes significative cleavage of the sensor equals 0.1 fM ($=10^{-16}$ M) or 0.003 U (Supplementary Fig. 4d). As a normal lethal assay has sensitivity in a range of 0.6–2.5 mouse LD50 Unit[42], we conclude that our assay is suitable for testing pharmacological preparation. While the cell-based assay developed here has thus the necessary sensitivity to serve a batch-release cell-based assay, further validation will be required to use it for such purposes.

The sensitivities measured here for BoNT/A and E are similar to those obtained by western blot cleavage assay in the same cellular model after 14 days of culture in a previous work of our group[30]. Interestingly, in this previous study and in this work, BoNT/E is more potent compared to A. This does not fit with previous experiments done on primary rodent cells[43] that indicated that BoNT/A was 30 times more potent than BoNT/E. But our findings match with the observations from the first clinical trial in humans (phase I study) in which it was shown that BoNT/E potency is at least equivalent or even greater compared to toxin A[44].

Finally, to prove we can use our system to test different steps of toxin intoxication, we tried to modulate toxin binding by a pre-exposure of our cells with gangliosides. As it was previously shown for iCell GABA neurons[26], we found that GT1b addition also improves BoNT/A potency in MNs (Fig. 4a). This observation differs from earlier findings which indicated that pre-exposure with gangliosides on iCell MNs did not impact their sensitivity to BoNT/A[28]. This discrepancy could be explained by the difference in protocols for gangliosides and toxin treatment. In the previous study, iCell MNs were pre-treated with 45 µM GT1b for 24 h prior to BoNT/A exposure while in our study, cells were pre-treated with 30 µM for 48 h prior to toxin exposure. Apart from these small differences in protocols, another important parameter to differentiate these two experiments is the readout which is employed to quantify the action of BoNTs. This suggests that the luminescent tool is more sensitive to small changes in BoNT cleavage compared to western blot. We also showed that our assay can be used to study toxin translocation. The blockage of this cellular process in our cells completely inhibited BoNT/A activity.

Nevertheless, the CB-assay presented here can be improved. One way to do it is to generate a stable hiPSC line expressing the SNAP-25 luminescent probe. From these stem cells, it would be possible to generate specialized neurons (MNs, glutamatergic neurons, sensory neurons) bankable as precursors. This approach would overcome the usage of lentivirus, limit the number of manipulations and reduce the time of experiment. All these improvements will facilitate the use of these engineered cells in a QC environment and/or for high-throughput screening.

## Methods

**Plasmids**. For BoNT/A detection in cell-free assays, the construct was cloned into pET28a vector with NocI/NotI and the insert was N-Nano-p25-C-Nano. Split NanoLuc sequence was kindly provided by Promega[33]. Overlap PCR was used to obtain full inserts and then ligation of the insert with cutted vector. The full a.a. sequence of this sensor is:

MVFTLEDFVGDWEQTAAYNLDQVLEQGGVSSLLQNLAVSVTPIQRIVRS-GENALKIDIHVIIPYEGLSADQMAQIEEVFKVVYPVDDHHFKVILPYGTLVI DGVTPNMLNYFGRPYEGIAVFDGKKITVTGTLWNGNKIIDERLITPDGSM LFRVTINSGSSGGGGSGGGGSSGGAQSARENEMDENLEQVSGIIGNLRHMA LDMGNEIDTQNRQIDRIMEKADSNKTRIDEANQRATKMLGSGGNSGSSGG GGSGGGGSSGVTGYRLFEEIL*

The primers for this construct are:
N-NanoFor: 5'-atatat cc ATGGTCTTCACACTCGAAGATTTC

C-NanoRev: 5'-atatat gcggccgc CAGAATCTCCTCGAACAGCC

P25For: 5'-atatat GAGCTCag GCCCGGGAAAATGAAATGG

P25Rev: 5'-atatat GAATTCccACCACTTCCCAGCATCTTTG

For BoNT/B detection in cell-free assays, the construct was N-Nano-vp1-C-Nano.

For BoNT/A detection in cell-based assays, the construct was designed using split NanoLuc, then firefly as internal controls. The internal control is located in front of SNAP-25 (full length). The vector was based on dual hSyn promotor vector[38] and BamHI/NotI were used for cloning. This cell-based sensor is named here "BoNT sensor4". The full a.a. sequence of BoNT sensor4 is

MVFTLEDFVGDWEQTAAYNLDQVLEQGGVSSLLQNLAVSVTPIQRIVRS-GENALKIDIHVIIPYEGLSADQMAQIEEVFKVVYPVDDHHFKVILPYGTL-VIDGVTPNMLNYFGRPYEGIAVFDGKKITVTGTLWNGNKIIDERLITPDGS-MLFRVTINSGSSGGGGSGGGGSSGGAQEDAKNIKKGPAPFYPLEDGTA-GEQLHKAMKRYALVPGTIAFTDAHIEVDITYAEYFEMSVRLAEAMK-RYGLNTNHRIVVCSENSLQFFMPVLGALFIGVAVAPANDIYNERELLNSM-GISQPTVVFVSKKGLQKILNVQKKLPIIQKIIIMDSKTDYQGFQSMYTFV-TSHLPPGFNEYDFVPESFDRDKTIALIMNSSGSTGLPKGVALPHRTACVRF-SHARDPIFGNQIIPDTAILSVVPFHHGFGMFTTLGYLICGFRVVLMYR-FEEELFLRSLQDYKIQSALLVPTLFSFFAKSTLIDKYDLSNLHEIASGGAPLSKEV-GEAVAKRFHLPGIRQGYGYGLTETTSAILITPEGDDKPGAVGKVVPFFEAKVVDL-DTGKTLGVNQRGELCVRGPMIMSGYVNNPEATNALIDKDGWLHSGDIAYW-DEDEHFFIVDRLKSLIKYKGYQVAPAELESILLQHPNIFDAGVAGLPDDDA-GELPAAVVVLEHGKTMTEKEIVDYVASQVTTAKKLRGGVVFVDEVPKGL-TGKLDARKIREILIKAKKGGKIAVGGGGSAEDADMRNELEEMQRRADQLA-DESLESTRRMLQLVEESKDAGIRTLVMLDEQGEQLERIEEGMDQINKDM-KEAEKNLTDLGKFCGLCVCPCNKLKSSDAYKKAWGNNQDGVVASQPARVV-DEREQMAISGGFIRRVTNDARENEMDENLEQVSGIIGNLRHMALDMG-NEIDTQNRQIDRIMEKADSNKTRIDEANQRATKMLGSGGNSGSSGGGGSG-GGGSSGVTGYRLFEEIL*

The primers for this construct are:

SalFireflyFor: 5'-atatat GAGCTCag GAAGATGCCAAAAACATTAAGAAGGG CCC

CFirefly-P25 overlap: 5'-CATTACGCATGTCTGCGTCCTCGGCGGAACCTC CGCCCCCCACGGCGATCTTGCCGC

NFirefly-P25 overlap: 5'-GCGGCAAGATCGCCGTGGGGGGCGGAGGTTCC GCCGAGGACGCAGACATGCGTAATG

EcoRIP25Rev: 5'-atatat GAATTCccACCACTTCCCAGCATCTTTGTTGC

BamHIN-NanoFor: 5'-atatat Ggatcc ATGGTCTTCACACTCGAAGATTTC

NotIC-NanoRev: 5'-atatat gcggccgc TTA CAGAATCTCCTCGAACAGCC.

## Lentiviral vector production
A replication-deficient lentiviral vector based on HIV virus was produced by Neo Biotech (CliniSciences, Nanterre, France) using the dual promotor vector expressing the polypeptide BoNT sensor4 as a lentiviral expression vector. The concentration of lentiviral vector particles was $1.3 \times 10^9$ TU/ml.

## His tag sensor proteins purification
Different constructs of sensor proteins were purified using IMAC. The His6 tag was cloned into pET28a vector at C terminal. BL21 containing plasmid of in vitro construct was inoculated overnight in LB medium with Kanamycin (50 µg/ml) and cultivated at 37 °C until O.D. around 0.6–0.8, then induced by 0.25 mM of IPTG (final concentration) for overnight at 20 °C. The cells are harvested by centrifuging at 4500 g, at 4 °C, 10 min.

After that, cells pellets were dissolved in 50 mM HEPES, 150 mM NaCl, pH 7.4 buffer and followed by sonication for 3 min. The supernatant of cell lysate was saved and mixed with 500 µl Ni2+ IMAC resin slurry for 1 h with rotation in 4 °C.

Later, the mixture was loaded into a column and the resin was washed with 5 ml of 20 mM Imidizole dissolved in 50 mM HEPES, 150 mM NaCl, pH 7.4 buffer.

Finally, the protein was eluted in four fractions and each fraction was 500 µl of 500 mM Imidizole in 50 mM HEPES, 150 mM NaCl, pH 7.4 buffer. The purified sensor protein was dialyzed into 1 l of 50 mM HEPES, pH 7.1 buffer at 4 °C and changed the buffer after 2 h and dialyzed overnight at 4 °C to perform the assay immediately.

The yield was ~500 µg for 200 ml of culture.

## BoNTs
For acellular assays and CB-assays using rodent neurons, isolated BoNT/A and BoNT/B1 (150 kDa) were purchased from Metabiologics (Madison, WI, USA). The activity of these toxins was reported to be $\sim 2 \times 10^8$ LD$_{50}$/mg by the vendor, which was not further validated in our laboratory. Toxins were stored in PBS (1 mg/ml) in −80 °C.

Recombinant research grade BoNT/A1 and E1-3 were manufactured from *E. coli* by Ipsen as previously described[44–47]. Inactive (endonegative) BoNT/1 (BoNT/A(0)) was produced by point mutations E224Q, H227Y in BoNT/A[38]. All BoNTs were purified and activated to more than 91% final purity. All recombinant toxins were activated with Lys-C and purified using protein liquid chromatography to more than 91%, as determined by SDS–polyacrylamide gel electrophoresis (PAGE). For assessing the effect of BoNT serotypes A and E, hiPSC-derived neurons were plated in a 96-well plate and exposed to the indicated doses of BoNT/A or BoNT/E in 100 µl of culture medium. For each experiment, each dose of BoNT was tested in triplicate. A negative control (medium without toxin) was always included.

## Acellular detection of BoNTs
Sensor protein concentration was estimated by SDS-PAGE with standard BSA as reference[30]. nM of sensor protein was prepared in 50 mM HEPES, 20 µM ZnCl2, 2 mM DTT, pH 7.1 buffer. The BoNT/A was diluted and added into sensor protein solution with a concentration series from 1 nM to 1 fM with dilution factor 10. After 4 and 24 h incubation at 37 °C, Nano-Glo substrate (Promega) was added to each sample with equal volume. The luminescent signal was measured in a plate reader. The assay was performed in duplicate. Negative controls were also designed and performed. More precisely, BoNT/A was mixed with N-Nano-p25-C-Nano sensor protein and the negative control was the mix of BoNT/A with the N-Nano-vp1-C-Nano sensor. On the other hand, BoNT/B was mixed with N-Nano-vp1-C-Nano sensor protein and the negative control was the mix of BoNT/B with the N-Nano-p25-C-Nano sensor protein.

## Rodent cell assay
Pregnant rats were purchased from Charles River. Primary cortical neurons were prepared from embryonic day (E) 18–19 pups. Dissected cortical tissue was digested in papain solution following the manufacturer's instructions (Worthington Biochemical, NJ). Cells were plated on poly-D-lysine coated 24-well plate and cultured in Neurobasal medium supplemented with B-27 (2%) and Glutamax (Invitrogen).

Lentivirus particles expressing the polypeptide BoNT sensor4 were added into 7-day neuron cells and the BoNT/A of a series concentration from 300 to 1 pM were added into 13-day neuron cells with triplicate. After 48 h challenging to toxin, the medium was removed, and a luciferase assay was performed as described below.

All animal studies were conducted at Boston Children's Hospital and approved under Institutional Animal Care and Use Committee (protocol numbers 18-10-3794R and 19-12-4065). Studies were performed in accordance with the guidelines of the Association for Assessment and Accreditation of Laboratory Animal Care International. Pregnant rats (Sprague Dawley strain) were purchased from Charles River.

## hiPSC-derived motor neurons detection assay of botulinum toxin
Frozen iCell MNs provided by FujiFilm Cellular Dynamics International (FCDI, Madison, WI, USA) were thawed and plated according to the provider instructions at the recommended density ($10^5$ cells/cm$^2$) into 96-well plates (TPP, Trasadingen, Switzerland or Greiner Bio-One International GmbH, Kremsmünster, Austria) in a volume of 100 µl per well. Cells were maintained in the medium provided by FCDI at 37 °C, 5% CO$_2$ for the indicated time. A 50–75% medium change was done every 2–3 days as recommended by suppliers. Cultures were observed regularly using an Olympus microscope (CKX53, Olympus Life Science Solutions, Waltham, MA, USA).

Forty-eight hours after plating, iCell MN were transduced with lentivirus particles expressing the polypeptide BoNT sensor4, at MOI of 16, in 40 µl of culture media provided by FCDI. Four to six hours after, the media was removed, and 100 µl of fresh medium was added. After 8 days of culture, cells were treated with BoNTs for 24 h, after which time the medium containing the toxin was removed and a luciferase assay was performed.

## Immunostaining
Eight days after plating, iCell MNs were fixed with 4% paraformaldehyde for 15 min at RT. Cultures were washed twice with PBS (Thermo Fisher Scientific) and then permeabilized and blocked with PBS containing 2% BSA (Sigma-Aldrich, Saint-Louis, MI, USA) and 0.1% triton (X100, Sigma-Aldrich) for at least 1 h at RT. Primary antibodies were incubated overnight at 4 °C in PBS 2% BSA, 0.1% triton. Cultures were then washed with PBS and stained with the corresponding secondary antibodies and DAPI for 1 h at RT and then washed with PBS. Primary antibodies used in this study were: Chicken Anti-Tubulin (Ab107216, Abcam Cambridge, United Kingdom; 1:1000), Goat Anti-Islet1 (GT15051, Neuromics Inc., Edina, MN, USA; 1:200). Secondary antibodies, Alexa 647 Donkey anti-Goat (A21447, Thermo Fisher Scientific), Alexa 594 Goat anti-Chicken (A11042, Thermo Fisher Scientific), all being used at 1:1000 dilution. Finally, cell cultures were washed in PBS.

Images were acquired at room temperature using the ImageXpress Micro Confocal High-Content Imaging System (Molecular Devices, San José, CA, USA) at ×20 magnification. Images were analyzed in MetaXpress software (Molecular Devices). A multi-wavelength cell scoring analysis was used to count DAPI+ and ISLET+ nuclei.

## Western blot
All samples were resolved by SDS-PAGE (Invitrogen, Carlsbad, CA, USA) and transferred onto nitrocellulose membranes. Membranes were blocked with 5% non-fat milk in 0.1% PBS-T (phosphate buffered saline with Tween-20) and probed with an anti-Firefly antibody (ab185924, Abcam, 1/5000) or anti-SNAP-25 antibody (S9684, Sigma-Aldrich, 1/2000). Immunoreactive bands were detected using horseradish peroxidase–conjugated secondary anti-rabbit (A6154, Sigma-Aldrich, 1/2000) antibodies and SuperSignal West Dura ECL substrate (Thermo Fisher Scientific). Bands were visualized on a Pxi4 imaging system using GeneSys image acquisition software (Syngene, Bangalore, Karnataka, India).

For SNAP-25 cleavage assay, intensities of the total form and cleaved form of SNAP-25 were measured with GeneTools (Philomath, OR, USA).

**Luciferase assay.** For rodent neurons assay, cells were lysed by adding 200 μl passive lysis buffer (Promega) and incubated at room temperature for 20 min. 50 μl of cell lysate was mixed with 50 μl firefly luciferase substrate (ONE-Glo(TM) EX reagent) and measured. Subsequently, 50 μl NanoLuc luciferase substrate (NanoDLR Stop & Glo(R) reagent) was added to the mix and measured again.

For hiPSCs derived neurons assay, after cell culture medium removing, 80 l of fresh DMEM w/o phenol red was added. The cells were incubated for 30 min at RT.

Luciferase activity (NanoLuc™ and Firefly luciferase) was assayed using the Nano-Glo® Dual-luciferase Reporter kit accordingly to the manufacturer's protocols (Promega) and using the multimode plate reader VICTOR® Nivo™ (Perkin Elmer, Waltham, MA, USA). With this protocol, luminescence was measured in less than 1 h following toxin treatment.

For GT1b assay, 6 days after thawing, GT1b (30 μM, Sigma-Aldrich) was added to the media provided by FCDI of previously transduced iCell MNs. Forty-eight hours later, this media was removed, and the cells were treated with rBoNT/A (1 pM). After 24 h the medium containing the toxin was removed and a luciferase assay was performed.

For ConA Assay, 8 days after thawing, transduced iCell MNs were treated with rBoNT/A (1 nM) in a medium provided by FCDI supplemented with 60 mM KCl for 10 min. Toxins were then washed off using PBS. Where appropriate, ConA (250 nM, Sigma-Aldrich) was added to the culture. Four hours later, a luciferase assay was performed.

For each cell lysate sample, both Firefly and NanoLuc™ luciferase signals, noted respectively FS and NS, were measured and the luminescence ratio, LR (LR = NS/FS) was calculated.

Then, for each sample, luminescence ratio was normalized with the mean luminescence ratio of the sample without toxin exposure (=MLR$_{w/o\ toxin}$) and data were expressed and graphed, as a percentage of this ratio as follows:

$$\text{Ratio percentage} = 100*(\text{LR}/\text{MLR}_{w/o\ toxin}) \tag{1}$$

Data were fitted to a four-parameter logistic equation and the negative logarithm of the EC50 (=pEC50) was calculated.

For one specific toxin concentration, BoNT sensor4 cleavage percentage is determined as follows:

$$\text{BoNT sensor4 cleavage}\,(\%) = 100*[1 - (\text{LR}/\text{LR}_{w/o\ toxin})] \tag{2}$$

Results shown are from four independent experiments in triplicates.

For acellular experiment, only NanoLuc™ luciferase signal, noted NS, was measured. The percentage of luminescent NanoLuc with toxin addition divided by the mean of NanoLuc signal of samples without toxin treatment (=MNS). This percentage was calculated as follows:

$$\text{Nanoluc percentage} = 100*(\text{NS}/\text{MNS}_{w/o\ tox}) \tag{3}$$

**Statistics and reproducibility.** All results are presented as mean ± SEM of $n$ independent experiments (specified in figure legends). Dose–response curves were fitted with a four-parameter logistic equation, and the pEC50 was calculated. All data processing and statistical tests were carried out using GraphPad Prism version 8 (GraphPad Software Inc., La Jolla, CA, USA). Statistical analysis indicates one or two-tailed $t$-test or ordinary one-way ANOVA multiple comparisons $P$ values, as indicated in figure legends. Significance is indicated by asterisks or NS, as follows: $*P < 0.05$, $**P < 0.01$, $***P < 0.001$; $****P < 0.0001$; NS: not significant $P > 0.05$.

**Reporting summary.** Further information on research design is available in the Nature Portfolio Reporting Summary linked to this article.

# Data availability
All data generated during this study are included in this published article (and its Supplementary Information files: Supplementary Material and Supplementary Data 1). Row data of all figures are shown in Supplementary Data 1. All other data are available from the corresponding authors upon reasonable request.

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

## Acknowledgements

This study was partially funded by Ipsen. This study was also supported by grants from the National Institute of Health (NIH) (R01NS080833 to M.D). M.D. holds the Investigator in the Pathogenesis of Infectious Disease award from the Burroughs Wellcome Fund.

## Author contributions

J.K., C.N. and M.D. conceived the project. C.N. and L.C. designed experiments. L.C. performed most experiments and received assistance from F.Y., S.R., J.D.D.L. and with data processing and analyses. Supervision: C.N. Writing—original draft: L.C., C.N. Review and editing: J.K., C.N., F.Y. and M.D. All authors read and approved the final manuscript.

## Competing interests

J.K., C.N., L.C., S.R. and J.D.D.L. are/were Ipsen employees. Harvard Medical School has filed a patent application covering the split NanoLuc-based assays for detecting botulinum neurotoxins, with F.Y. and M.D. as co-inventors.

## Ethical approval

The methods were performed in accordance with relevant guidelines and regulations and approved by The National Agency for the Safety of Medicines and Health Products (ANSM).
