## [Peer Review File · Communications Biology]

Reviewers' comments:

Reviewer #1 (Remarks to the Author):

The authors describe a new cell-based potency assay for botulinum toxin that could potentially replace the still frequently used mouse lethality assay. They use a split luciferase that is transiently expressed as transgene in iPS cell-derived motor neurons as artificial substrate for the small subunit's protease activity. By competing or enhancing the heavy chain-mediated uptake of the toxin, they convincingly show that the test records toxin binding and uptake apart from substrate cleavage. The results are clearly presented and limitations of the current setup of the assay are discussed adequately.

However, one major shortcoming should be addressed more carefully: The assay determines the decrease of split nanoluc luciferase activity. The transgene contains a firefly luciferase as internal standard to correct for transfection efficacy. The BoNT activity is displayed as result of a multi-step normalization. (1) It remains elusive, why the authors used different approaches for calculation/normalization in the different experiments; please comment. (2) The description of the calculation in the methods section is somewhat insufficient, because not all variables of the calculation are clearly defined. This needs to be improved. (3) More importantly, original luciferase data should be provided along with example calculations in the supplementary data to give an idea of how much the nanoluc activity actually decreases upon BoNT treatment and how stable the reference luciferase firefly is between different measurements. In addition, the authors should indicate how much Rw toxin MAX varied between different experiments.

Figure 2: To prove that transduction does not affect differentiation of iPSC into motor neurons, the authors actually would need to show that the fraction of ILET1-positive cells expressing luciferase is similar to the fraction of ILET1-positive cells in non-transfected cultures.

Minor:

The discussion of assays using fluorescent probes should be extended. The statement that the use of fluorescent probes requires complicated equipment is not very scientific. FRET-based probes follow the exact same principle as the split luciferase and fluorescence or FRET measurements can be performed in the exact same micro titer device that is used for the luciferase assay.

Line 473 Discussion Reference 28 used a Gaussia luciferase, not firefly.

Line 506 "compared" not "compare"

Reviewer #2 (Remarks to the Author):

The manuscript describes the development of a cell-based assay for botulinum toxins. The researchers make use of motor neurons derived from hiPSC, and antibody-free luminescent detection system based on split nanoluc luciferase. The concept of the assay is interesting as the assay is sensitive and the system can potentially be adjusted to all BoNT serotypes. However, the manuscript requires major revisions before publication. Comments are detailed below.

Major comments

1. The manuscript requires a thorough linguistic editing review.
2. Details regarding the principles of the nanoluc assay should be added to either the introduction or the results sections.
3. Lines 221-231 – the subtype of the toxins should be detailed. BoNT/B is not mentioned. References 23-25 do not provide enough details about the production of recombinant BoNTs. Moreover, details regarding BoNT purification and activation should be included. The specific activity of the toxins used

should be included and the method used to determine it.

4. Line 291 – the in vitro assay (figure 1) does not include Firefly. However, the R_i is defined in line 286 as nanoluc/Firefly. The author should explain their calculation precisely.
5. Line 331 – there is no scientific explanation to include the SV2C sequence in the in vitro experiment. The conclusion that it functions in this assay as a toxin receptor is unreasonable. The improved EC50 with this substrate may be associated to the length of the linker between the nanoluc parts, and appropriate control to support this function should be included. Additionally, the SV2C sequence was not included in the main cell-based sensor4 construct. The use of the construct with SV2C should not be included if not scientifically justified.
6. Figure 1 C and D – presenting a single point in a continuous graph (for BoNT/B in figure 1C and for BoNT/A in figure 1D) is not acceptable.
7. Figure 2A – the scheme includes BoNT sensor1 while the legend refers to sensor4.
8. Figure 2B – the figure shows that as the cleavage of SNAP25 is reduced with increasing BoNT concentration. Please clarify.
9. Figure 2C- the ratio percentage should decrease with increasing BoNT concentration, as presented in figure 1 C+D. However, this ratio increases in figure 2C with increasing BoNT concentration.
10. Figure 2B – the graph presented in the figure is based on three replicates of a Western blot analysis. The raw Western blot figures should be included in the supplementary file.
11. Lines 480-481 – This main result is not presented in the results section. Please explain how the value of 40 fg/ml was translated to 272 fM. Assuming a toxin molecular weight of 150,000 g/mol, a concentration of 272 fM equals 40 pg/ml.

Minor comments

1. All over the manuscript there are inconsistencies with the use of abbreviations. For example, in line 39 SNAP-25 is written with hyphen and in line 42 it is written without.
2. The abbreviations list does not include the full explanation or the abbreviated form for all terms.
3. Line 45 – the use of the word discriminate is inaccurate. For example, the assay cannot distinguish between BoNT/A and BoNT/E as both toxins cleave SNAP25.
4. Lines 84-93 – the intoxication mechanism steps should be rewritten into a combined paragraph.
5. Line 94 – Accurate and reliable test is relevant to all pharmaceuticals and is not limited to biologic and toxic preparations. The sentence should be corrected.
6. Line 97 – the sentence should be rewritten to accurately define LD50.
7. Line 106 – in this sentence the authors mention that several assays were approved by the FDA or other regulatory agencies. The authorization approval of the assays in references 14 and 15 are not familiar. Please elaborate on the regulatory agencies that approved the assays mentioned in references 14-15.
- 8.
9. Line 138 – the title should be in plural form.
10. Line 139 - please indicate the name of the cell line.
11. Line 144 – which type of VAMP was used – species and isotype.
12. Line 151 – the authors should provide details about the cell line and viruses used.
13. Line 158 – the word correct should be deleted.
14. Line 159 – the authors should add the medium used.
15. Line 167 – the authors should add the yield of protein expression.
16. Line 164 – which column was used.
17. Line 181 – the title should be rephrased.
18. Line 185 - the authors should add the names of the media used and their ingredients.
19. Line 190 – $100 \times 10^3 = 1 \times 10^5$
20. Line 192 - the authors should add the names of the media used and their ingredients.
21. Line 193 – the meaning of the word realized is not clear.
22. Line 213 – this section should be included under the immunostaining section.
23. Lines 215-216 – the magnification scale should be corrected (it cannot be X10).
24. Line 225 – the letter c in E. coli should not be capital.

25. Line 237 – the authors should add the medium used.
26. Line 257 – the authors should add the medium used.
27. Line 264 – the authors should add the medium used.
28. Lines 243 and 256 – the GT1B and ConA assays should be included under the luciferase assay.
29. Line 286 – the letter I in the Ri term should be explained.
30. Line 289 – “rodent experiment” should be rephrased to “rodent cells experiment”.
31. Line 320 –The first line should clearly indicate that the assay in this section is acellular.
32. Line 357 – The authors should explain how Firefly luciferase serves as an internal control.
33. Line 360 – the schematic representation of sensor4 should be presented in the manuscript and not in the supplementary file.
34. Line 388 – the values of pEC50 cannot be negative.
35. Line 410 – the authors should explain the reason for using different sources of toxins – recombinant toxins (Ipsen) or native toxins (Metabionics).
36. Line 412 – the authors should explain how they know the cleavage of SNAP25 is complete.
37. Line 419 – the term “mechanistic question” is general. The title needs to be rephrased.
38. Line 435 – not all intoxication steps are supported by the data. For example, binding to the SV2C protein receptor.
39. Lines 456-459 – the paragraph needs to be rewritten to include the background referring the time issue in iCell MN.
40. Line 466 – the conclusion that the assay is cheaper than ELISA is not convincing. The costs of working with hiPSC are relatively higher than working with cell lines. Additionally, the costs consideration should be added to the price of the nanoluc and firefly kits.
41. Line 499 – according to the suggested explanation, the authors should elaborate the possible differences between natural and recombinant BoNT that may clarify the discrepancy with reference 17.
42. Line 513 – the use of three dots in the parenthesis should be avoided.
43. Figure 2B – the digits in the x axis are not below the ticks.
44. Figure 4 Line 758 – it is not clear whether ConA was co-administered with the toxin or not.
45. Supplementary figures 2 and 4 – Supp. Figure 2 blot data is duplicated in Supp. Figure 4. The blot should be presented only once and referred accordingly.

Reviewer #3 (Remarks to the Author):

This manuscript describes an improvement of a previously described hiPSC derived motor neurons assay for detection of BoNT activity, using a previously described technology as a new read-out method, as previously proposed or described in other publications. Nevertheless, the development is exciting for pharmaceutical companies looking for potency assays for their product. It is less useful or feasible for research applications, although design of a novel stem cell line as proposed in the last section of the discussion would be an exciting development for research applications as well. Overall, the description of the study is somewhat disjointed, especially in methods section, making reading the paper rather laborious. In addition, the methods lack important detail required to reproduce results, and the entire manuscript suffers from incomplete literature citations. The wording in the manuscript is often vague, and several experiments are lacking to support the conclusions as detailed below:

1. Lines 40-41: Most in vitro tests ... This statement is inaccurate. First of all, there's a multitude of in vitro tests other than cell-based assays, and secondly, these days most investigators using cell based assays use stem cell derived neurons or primary neurons. As for pharmaceutical companies, one uses an immortalized cell line, another uses hiPSC derived neurons, others are still developing their assay. Finally, primary rodent cells are actually very accurate, and the main limitation with immortalized cell lines is lack of sensitivity.

2. Line 75-76: What exactly do the authors mean by BoNT-like toxins? Neither BoNT/X nor BoNT/En have been shown to be toxic, and certainly neither is toxic to vertebrates like BoNTs. Both are structurally similar, have similar endopeptidase functions, and are genetic homologs. As is BoNT/Wo and BoNT/Cp1-3.
3. Line 81: Ref 6 is appropriate for the part of this sentence, but the main focus is new developments of pharmaceutical BoNTs. There are many extensive reviews on clinical use of BoNTs for muscle dysfunctions and other indications, please add a general review here.
4. Line 89: please cite original literature or review focused on ganglioside binding
5. Line 90: Critical literature is missing. Please either cite all original literature including crystal structure of toxin-receptor complexes, or cite a recent comprehensive review focused on protein receptor binding.
6. Lines 112-116: this text and citation is misleading, as this test is not used for potency testing of pharmaceutical BoNTs. Even though BioSentinal has now developed an assay based on this technology, it is using a different cell line and possibly a different substrate.
7. Line 117-120: This has been reviewed and discussed extensively. Please cite reference: Pellett S.Curr Top Microbiol Immunol. 2013;364:257-85.
8. Line 123: this work does not describe development of a new assay, but rather improvement of a current assay with a new endpoint using a reporter, which has previously been suggested (see ref 15 and Pellett S.Curr Top Microbiol Immunol. 2013;364:257-85.)
9. Lines 125-126: Please describe whether other functional enzyme assays, and specifically cell based assays, have been developed using the split-Nanoluc luciferase technology.
10. Lines 130-135: This sentence should cite previous publications on assays using the exact same hiPSC derived neurons as are used in this study, and describing the same advantages just with a different endpoint. The use of hiPSC derived motorneurons is not new, and great sensitivity has been demonstrated previously.
11. Lines 139-149: Please be more specific on the design of constructs. Please provide source sequence accessions used, primer sequences, more detail on the internal control. etc. It is completely unclear what BoNT sensor 4 is from this description alone – please refer to suppl. Fig.
12. Lines 151-154: Please indicate the exact viruses used.
13. Line 167-168: Please provide more detail on dialysis – buffer volumes, number of changes, duration, temperature
14. Line 177: Please describe the negative controls
15. Lines 181-185: The section title of rodent detection implies in vivo assays. In addition, much more detail is required here. How were cells prepared, from which animal species, how were the cells cultured, how were the cells exposed to lentivirus, etc.
16. Line 218-219: This last sentence seems unnecessary
17. Line 222-223: Please provide more information on the purchased toxins. Were they complex or 150 kDa purified BoNT? What was the specific activity? Was activity confirmed upon receipt? How were they stored?
18. Line 227: What do the authors mean by the statement 'All BoNTs were activated to more than 91% final purity'? How were the toxins activated?
19. Line 257: Please provide the source of GT1b
20. Lines 276-277: Incomplete sentence
21. Lines 322-326 and Fig 1A: The construct of the split luciferase is unclear. Based on the text and figure legend, it would almost appear that the authors describe the SNARE target to be between the N and C terminal fragments of luciferase. Even though the figure shows the SNARE target is outside the luciferase gene, the circular design is confusing. Would it be possible to re-design the figure to show the fusion protein in a more descriptive manner, not as a circle? Please also change the text to better describe the basis of the split luciferase assay. Why is the linker to the N-terminal luciferase longer, and so much longer than the SNARE target itself? The entire sequence of the construct should be provided in some manner, or deposited.
22. Line 326-327: 'The remaining luciferase activity can be inversely 327 correlated with the toxin activity.' At this point in the ms, this should be phrased as a hypothesis.
23. Line 328: Please replace several with the number of constructs designed

24. Line 331: Please explain why SV2C was added, since it is a receptor for the HC not the LC of BoNT
25. Line 344: please add 'in an in vitro assay'
26. Lines 335-346: It should be clearly stated that this section describes an in vitro assay using reduced BoNTs
27. Figure 1c: the labels in the figure are quite small and difficult to see. Please increase the size of the labels. The figure legend states the assay was performed in duplicate, but one curve seems to show standard deviations. Please either show both curves for the duplicate assays or one representative curve if they are overlapping. In addition, please specify what duplicates refers to, were two independent assays conducted, two independent toxin dilution series, or the same dilution series in the same assay done in duplicate on the plate.
28. Figure 2: please increase font
29. Lines 363-365: As a common challenge of transfecting neurons with lentiviral vectors is cell death, please add data demonstrating cell viability and percentage of transfected cells.
30. Figure 2B: It is unclear what is shown in the graph. The Y-axis is labeled as percent SNAP-25 cleavage, but based on the description this was not directly measured.
31. Figure 2C: Similarly, the Y axis in this figure is labeled as ratio percentage, and while I can figure out what this means by flipping back and forth through the manuscript sections, it should be obvious from the figure alone. Please clarify labels and figure legend.
32. Line 371: Ref 26 is just an abstract, not a peer reviewed publication. The first peer reviewed publication showing sensitivity of hiPSC derived motor neurons to BoNTs is ref 17, which should be cited here. Another reference that should be cited is
Analysis of Motor Neurons Differentiated from Human Induced Pluripotent Stem Cells for the Use in Cell-Based Botulinum Neurotoxin Activity Assays. Schenke M, Schjeide BM, Püschel GP, Seeger B. *Toxins (Basel)*. 2020 Apr 25;12(5):276. doi: 10.3390/toxins12050276.
33. Line 374: 'the majority' is a vague term. Please be more specific.
34. Figure 2D: Why does GFP stain in a punctate pattern similar to DAPI or ISLET1? Wouldn't the reporter be expected to be cytosolic or membrane bound?
35. Lines 406-407: Why was transduction done at 48 h after plating, and the assay 6 days later?
36. Please define pEC50
37. Lines 414-415: Please also provide values showing the EC50 in Units of toxin activity, since this is a functional assay that measures toxin activity. Without this, the data cannot be compared to other literature.
38. Figure 3: An assay showing endogenous SNAP-25 cleavage by Western blot or ICC should be included for direct comparison, so that the sensitivity of this new endpoint can be evaluated.
39. Lines 426-427: 'increased' is vague. Please be more specific.
40. Figure 4: please increase label size
41. Line 424, ref 14: This was done using a different cell type.
42. Line 427: Without quantifying the extracellular GT1b with and without external addition, these data do not show mechanisms, especially with only a 10% difference in SNAP-25 cleavage at only one toxin concentration.
43. Lines 435-437: These conclusions are not supported by the two assays shown here. However, all steps of intoxication are required in a cell based assay by definition.
44. Line 445: Please change 'new assay' to 'a new endpoint for an existing assay' or 'an improvement on the hiPSC cell based assay'.
45. Line 450: please add citations for peer reviewed publications previously describing use of the exact same cells used here for the same purpose.
46. Line 451: delete 'new'
47. Line 454, ref 17: the ref is appropriate but please see earlier comments on additional ref
48. Line 461: "moreover" seems misplaced here. The luminescent read-out for this specific assay is the only novel aspect of this study.
49. Line 463: Why do the authors focus only on this ELISA read-out? There's other read-outs described including ICC, Western blot, other ELISA citations, neurotransmitter release...
50. Lines 473-478: This reads more like a company defending its product than scientific analysis of benefits and limitations. Please re-write to highlight the unique aspects of each read-out, i.e. serotype

specificity, versatility, specificity of endpoint, etc

51. Line 491: This conclusion is too broad. The authors specifically show greater sensitivity of one human motor neurons cell model to BoNT/E. This does not mean that human neurons in general are more sensitive to BoNT/E than to BoNT/A. – please see ref 17

52. Line 493-506: This is a long discussion comparing results from one citation that says 'data not shown' and states no difference was noted, to results from this study which used only one toxin concentration and noted a modest difference only. Unless these data are repeated with a dose response curve, conclusions can only be limited.

Point by Point responses to reviewers

Manuscript number: COMMSBIO-21-1802

Manuscript title: Split luciferase-based assay to detect botulinum neurotoxins using human motor neurons derived from induced pluripotent stem cell.

Reviewer #1 (Remarks to the Author):

The authors describe a new cell-based potency assay for botulinum toxin that could potentially replace the still frequently used mouse lethality assay. They use a split luciferase that is transiently expressed as transgene in iPS cell derived motor neurons as artificial substrate for the small subunit's protease activity. By competing or enhancing the heavy chain-mediated uptake of the toxin, they convincingly show that the test records toxin binding and uptake apart from substrate cleavage. The results are clearly presented and limitations of the current setup of the assay are discussed adequately.

However, one major shortcoming should be addressed more carefully: The assay determines the decrease of split nanoluc luciferase activity. The transgene contains a firefly luciferase as internal standard to correct for transfection efficacy. The BoNT activity is displayed as result of a multi-step normalization.

(1) It remains elusive, why the authors used different approaches for calculation/normalization in the different experiments; please comment

The method of calculation was homogenized between Rodent and human model. For each sample, the percentage of the luminescent ratio signals normalized by the luminescent ratio of sample non treated with toxin was calculated and plotted; please see the Material & Methods section *line 323 to 341*.

(2) The description of the calculation in the methods section is somewhat insufficient, because not all variables of the calculation are clearly defined. This needs to be improved

Description of calculation has been improved; please see *line 323 to 341* in the Material & Methods section.

(3) More importantly, original luciferase data should be provided along with example calculations in the supplementary data to give an idea of how much the nanoluc activity actually decreases upon BoNT treatment and how stable the reference luciferase firefly is between different measurements. In addition, the authors should indicate how much R_w toxin MAX varied between different experiments

An example of calculation has been implemented in the revised manuscript, please see supplementary figure 5. Also, a graph of luminescent signals (Firefly and Nanoluc) obtained after BoNT/E treatment has been incorporated into the supplementary figure 4 : The graph in the supplementary figure 4B shows mean of 4 independent experiments describing the Firefly signal and the decrease of the Nanoluc signal; the graph in supplementary figure 4C shows the percentage of the luminescent ratio when cells were treated with the highest dose of BoNT A and E respectively.

"Figure 2: To prove that transduction does not affect differentiation of iPSC into motor neurons, the authors actually would need to show that the fraction of ILET1-positive cells expressing luciferase is similar to the fraction of ILET1-positive cells in non-transfected cultures"

The analysis of Islet1 and Firefly expression has been conducted by immunofluorescence and results have been incorporated in Figure 2 (2D and 2E). With this

analysis, we showed that the ISLET+cells expressing Firefly is similar to the ISLET + cells in non-transfected cultures. Text has been modified accordingly (*line 434 to 439*).

The discussion of assays using fluorescent probes should be extended. The statement that the use of fluorescent probes requires complicated equipment is not very scientific. FRET-based probes follow the exact same principle as the split luciferase and fluorescence or FRET measurements can be performed in the exact same micro titer device that is used for the luciferase assay

We agree with the comment and the discussion was rewritten and improved. Please see the discussion section from *line 516*.

Line 473 Discussion Reference 28 used a Gaussia luciferase, not firefly

We corrected this mistake and indicated that it is a Gaussia luciferase. Please see the discussion section from *lines 523-526*.

Line 506 "compared" not "compare

Text was corrected. Please see the discussion section from *line 563*.

Reviewer #2 (Remarks to the Author):

The manuscript describes the development of a cell-based assay for botulinum toxins. The researchers make use of motor neurons derived from hiPSC, and antibody-free luminescent detection system based on split nanoluc luciferase. The concept of the assay is interesting as the assay is sensitive and the system can potentially be adjusted to all BoNT serotypes. However, the manuscript requires major revisions before publication. Comments are detailed below.

Major comments

1. The manuscript requires a thorough linguistic editing review

Manuscript has been reviewed to allow linguistic improvement

2. Details regarding the principles of the nanoluc assay should be added to either the introduction or the results sections

The manuscript has been improved to address this point. A first description of the Nanoluc sensor is now given in the last part of introduction and additional explanations are given in the first paragraph of the results section. Please see the introduction section from *line 125-128* and result section from *line 360-364*.

BoNT/B is BoNT/B1 purchased from Metabionics (catalog number #210).

3. Lines 221-331 - the subtype of the toxins should be detailed. BoNT/B is not mentioned. References 23-25 do not provide enough details about the production of recombinant BoNTs. Moreover, details regarding BoNT purification and activation should be included. The specific activity of the toxins used should be included and the method used to determine it

Details on the different toxins' subtypes used in the study have been added in the material and method section and the Reference "*Elliott et al., Sci. Adv. 2019; 5*" has been added to provide information on the procedures for recombinant BoNT production and analysis.

4. Line 291 – the in vitro assay (figure 1) does not include Firefly. However, the Ri is defined in line 286 as

nanoluc/Firefly. The author should explain their calculation precisely.

The figure 1 explains the general concept of the split Nanoluc sensor and presents results obtained with acellular assays. Sensors used for these acellular assays do not contain Firefly.

The Firefly is present in the sensors used for cellular assays as it is used as transduction control.

The way we calculated the Nanoluc percentage for the figure 1 is described in *lines 337-341* of material and method.

5. Line 331 – there is no scientific explanation to include the SV2C sequence in the in vitro experiment. The conclusion that it functions in this assay as a toxin receptor is unreasonable. The improved EC50 with this substrate may be associated to the length of the linker between the nanoluc parts, and appropriate control to support this function should be included. Additionally, the SV2C sequence was not included in the main cell-based sensor4 construct. The use of the construct with SV2C should not be included if not scientifically justified.

We agree the inclusion of the SV2C sequence in the acellular experiment is not justified; these data have been removed.

6. Figure 1 C and D – presenting a single point in a continuous graph (for BoNT/B in figure 1C and for BoNT/A in figure 1D) is not acceptable

The results for BoNT/A in the previous manuscript figure 1D and for BoNT/B in the previous figure 1C have been removed from the Figure 1 and plotted separately in Supplementary Figure 1.

7. Figure 2A – the scheme includes BoNT sensor1 while the legend refers to sensor4.

This scheme and its legend were corrected. The scheme was moved to Supplementary figure 2. Figure 2 was modified.

8. Figure 2B – the figure shows that as the cleavage of SNAP25 is reduced with increasing BoNT concentration. Please clarify.

The legend of the Y axis of figure 2B was unfortunately switched and we apologize for this mistake. The legend of the figure 2B was corrected.

Figure 2B shows the decreasing ratio percentage when we add increasing amount of toxin.

9. Figure 2C- the ratio percentage should decrease with increasing BoNT concentration, as presented in figure 1 C+D. However, this ratio increases in figure 2C with increasing BoNT concentration

The legend of the Y axis of figure 2C was unfortunately switched and we apologize for this mistake. The legend of the figure 2C was corrected.

Figure 2C shows the decreasing ratio percentage when we add increasing amount of toxin.

10. Figure 2B – the graph presented in the figure is based on three replicates of a Western blot analysis. The raw Western blot figures should be included in the supplementary file.

Blots were added in Supplementary Figure 3.

11. Lines 480-481 – This main result is not presented in the results section. Please explain how the value of 40 fg/ml was translated to 272 fM. Assuming a toxin molecular weight of 150,000 g/mol, a concentration of 272 fM equals 40 pg/ml.

We agree that 272 fM equals 40pg/ml and we apologize for the miscalculation. The text has been corrected, please see *line 533-534*.

Minor comments

1. All over the manuscript there are inconsistencies with the use of abbreviations. For example, in line 39 SNAP-25 is written with hyphen and in line 42 it is written without.

We write SNAP-25 with the hyphen and stick to it all along the manuscript.

2. The abbreviations list does not include the full explanation or the abbreviated form for all terms. The abbreviation list was extended.

3. Line 45 – the use of the word discriminate is inaccurate. For example, the assay cannot distinguish between BoNT/A and BoNT/E as both toxins cleave SNAP25

The word “discriminated” was removed from the abstract.

4. Lines 84-93 – the intoxication mechanism steps should be rewritten into a combined paragraph The intoxication mechanism paragraph was rewritten into a combined paragraph. Please see the introduction section from *line 86 to 95*.

5. Line 94 – Accurate and reliable test is relevant to all pharmaceuticals and is not limited to biologic and toxic preparations. The sentence should be corrected.

The sentence was corrected. Please see the introduction section from *line 96 to 97*.

6. Line 97 – the sentence should be rewritten to accurately define LD50.

Text was rewritten. Please see the introduction section from *line 98 to 100*.

7. Line 106 – in this sentence the authors mention that several assays were approved by the FDA or other regulatory agencies. The authorization approval of the assays in references 14 and 15 are not familiar. Please elaborate on the regulatory agencies that approved the assays mentioned in references 14-15

This paragraph was rewritten. Please see the introduction section from *line 104 to 107*.

8. Line 138 – the title should be in plural form.

This title was corrected, please see line 136

9. Line 139 - please indicate the name of the cell line.

The name of the cell line was indicated, please see *line 243-244*.

10. Line 144 – which type of VAMP was used – species and isotype.

It's human VAMP1. Name has been changed to “hVAMP1”, see *line 152*

11. Line 151 – the authors should provide details about the cell line and viruses used.

Details on the virus types and concentration were added, please see line 189-192.

12. Line 158 – the word correct should be deleted

The word correct was deleted.

13. Line 159 – the authors should add the medium used

The text was changed. Please see *line 197*.

14. Line 167 – the authors should add the yield of protein expression

The text was change. Please see *line 210*.

15. Line 164 – which column was used

The text was changed. Please see *line 203*.

16. Line 181 – the title should be rephrased
The title was rephrased. Please see *line 242*.
17. Line 185 - the authors should add the names of the media used and their ingredients
The name of the different media used were added in the method section. Please see *line 246-247*.
18. Line 190 – $100 \times 10^3 = 1 \times 10^5$
The text was corrected. Please see *line 256*.
19. Line 192 - the authors should add the names of the media used and their ingredients
iPSC- derived Motor neurons and media were provided by CDI, the exact composition of the media is not known. This is the reason why we mentioned "media provided by CDI" in the text.
20. Line 193 – the meaning of the word realized is not clear
The text was rewritten. Please see *line 259*.
21. Line 213 – this section should be included under the immunostaining section
This section has been added to the immunostaining section. Please see *lines 282 -285*.
22. Lines 215-216 – the magnification scale should be corrected (it cannot be X10).
The magnification was corrected. The images were taken at 20X. Please see *line 283*.
23. Line 225 – the letter c in E. coli should not be capital
The name was corrected. Please see *line 217*.
24. Line 237 – the authors should add the medium used
iPSC- derived Motor neurons and media were provided by CDI, the exact composition of the media is not known. This is the reason why we mentioned "media provided by CDI" in the text. Please see *line 258-259*.
25. Line 257 – the authors should add the medium used.
iPSC- derived Motor neurons and media were provided by CDI, the exact composition of the media is not known. This is the reason why we mentioned "media provided by CDI" in the text. Please see *line 258-259*.
26. Line 264 – the authors should add the medium used
iPSC- derived Motor neurons and media were provided by CDI, the exact composition of the media is not known. This is the reason why we mentioned "media provided by CDI" in the text. Please see *line 258-259*.
27. Lines 243 and 256 – the GT1B and ConA assays should be included under the luciferase assay
the GT1B and ConA assays was included under the luciferase assay. Please see the material and method section section *line 313 to 321*.
28. Line 286 – the letter l in the Ri term should be explained

The calculation section in the Material & Methods section was completely rewritten and this abbreviation was removed; please see from *lines 324*.

29. Line 289 – “rodent experiment” should be rephrased to “rodent cells experiment”

As the method of calculation was homogenized between cells experiment (rodent and humancells), the expression rodent experiment was removed. Please see the material and method section section *line 323 to 331*.

30. Line 320 –The first line should clearly indicate that the assay in this section is acellular

There is now a clear indication that the assay used in this section is acellular, see lines 365 and 366.

31. Line 357 – The authors should explain how Firefly luciferase serves as an internal control

Firefly is expressed in the same polypeptide and can be detected even when the Nanoluc luciferase is cleaved; so this luciferase can be used to control transfection rate and protein expression levels.

32. Line 360 – the schematic representation of sensor4 should be presented in the manuscript and not in the supplementary file

Schematic representation of BoNT sensor 4 has been incorporated to the Figure 2, please see Figure 2 A.

33. Line 388 – the values of pEC50 cannot be negative

We apologize for that mistake. This was corrected in the text. Please see *line 424*.

34. Line 410 – the authors should explain the reason for using different sources of toxins – recombinant toxins (Ipsen) or native toxins (Metabiologics).

The work presented in that study was conducted at two separate laboratories; therefore two different sources of toxins were used.

35. Line 412 – the authors should explain how they know the cleavage of SNAP25 is complete

To confirm the cleavage of SNAP-25 is complete ; Western blot was performed after treatment with the highest dose (9 nM) of both toxins and we showed that the sensor was completely cleaved.

The results of that analysis have been added to the Supplementary Figure 4, please see Figure 4 A.

36. Line 419 – the term “mechanistic question” is general. The title needs to be rephrased

The title was modified, please see line 463.

37. Line 435 – not all intoxication steps are supported by the data. For example, binding to the SV2C protein receptor.

The text was corrected, please see the conclusion of the last paragraph on the result section. Please see *line 482-484*.

38. Lines 456-459 – the paragraph needs to be rewritten to include the background referring the time issue in iCell MN.

This paragraph was modified accordingly, please see the paragraph from line 498 to 508.

39. Line 466 – the conclusion that the assay is cheaper than ELISA is not convincing. The costs of working with hiPSC are relatively higher than working with cell lines. Additionally, the costs consideration should be added to the price of the nanoluc and firefly kits.

We agree that the cost is not a solid argument; compared to the ELISA assay. Our assay has the advantage to do not need a specific antibody. The text has been modified accordingly. Please see the paragraph from line 510 to 515.

40. Line 499 – according to the suggested explanation, the authors should elaborate the possible differences between natural and recombinant BoNT that may clarify the discrepancy with reference 17.

The different natures of toxin don't seem to be a good explanation of this discrepancy as it was shown that both natural and recombinant BoNT/A manufactured by IPSEN show similar potency in vivo (Périer C et al. 2021, Recombinant botulinum neurotoxin serotype A1 in vivo characterization). The differences observed could be explained by the different experimental protocols.

41. Line 513 – the use of three dots in the parenthesis should be avoided

The 3 dots were suppressed.

42. Figure 2B – the digits in the x axis are not below the ticks

Figure 2B was corrected.

43. Figure 4 Line 758 – it is not clear whether ConA was co-administered with the toxin or not

ConA was not co-administrated. ConA was added 10 min after BoNT treatment (the media containing the toxin was removed and a fresh one was added with ConA).The figure 4 has been corrected.

44. Supplementary figures 2 and 4 – Supp. Figure 2 blot data is duplicated in Supp. Figure 4.

The blot should be presented only once and referred accordingly

Figures have been corrected and the blot was shown only once, please see supplementary figure 2C.

Reviewer #3 (Remarks to the Author):

This manuscript describes an improvement of a previously described hiPSC derived motor neurons assay for detection of BoNT activity, using a previously described technology as a new read-out method, as previously proposed or described in other publications. Nevertheless, the development is exciting for pharmaceutical companies looking for potency assays for their product. It is less useful or feasible for research applications, although design of a novel stem cell line as proposed in the last section of the discussion would be an exciting development for research applications as well. Overall, the description of the study is somewhat disjointed, especially in methods section, making reading the paper rather laborious. In addition, the methods lack important detail required to reproduce results, and the entire manuscript suffers from incomplete literature citations.

The wording in the manuscript is often vague, and several experiments are lacking to support the conclusions as detailed below:

1. Lines 40-41: Most in vitro tests ... This statement is inaccurate. First of all, there's a multitude of in vitro tests other than cell-based assays, and secondly, these days most investigators using

cell based assays use stem cell derived neurons or primary neurons. As for pharmaceutical companies, one uses an immortalized cell line, another uses hiPSC derived neurons, others are still developing their assay. Finally, primary rodent cells are actually very accurate, and the main limitation with immortalized cell lines is lack of sensitivity.

The abstract was modified, and the mention “most in vitro tests” removed. Please see line 43.

2. Line 75-76: What exactly do the authors mean by BoNT-like toxins? Neither BoNT/X not BoNT/En have been shown to be toxic, and certainly neither is toxic to vertebrates like BoNTs. Both are structurally similar, have similar endopeptidase functions, and are genetic homologs. As is BoNT/Wo and BoNT/Cp1-3

As these toxins are less relevant to the main topic of the manuscript, we deleted this sentence and references.

3. Line 81: Ref 6 is appropriate for the part of this sentence, but the main focus is new developments of pharmaceutical BoNTs. There are many extensive reviews on clinical use of BoNTs for muscle dysfunctions and other indications, please add a general review here.

The reference [7] “Dressler D (2012) Clinical applications of botulinum toxin. In *Curr Opin Microbiol*, pp. 325–36, England” was added to the manuscript to illustrate the point. Please see line 82 of the new manuscript.

4. Line 89: please cite original literature or review focused on ganglioside binding

These two references :[8] “Rummel, A. (2016). Two feet on the membrane: uptake of clostridial neurotoxins, in *Uptake and Trafficking of protein toxins*. Editor H. Barth (Cham: Springer International Publishing), 1–37. doi:10.1007/82_2016_48” ; [9] “Hamark, C., Berntsson, R. P.-A., Masuyer, G., Henriksson, L. M., Gustafsson, R., Stenmark, P., et al. (2017). Glycans confer specificity to the recognition of ganglioside receptors by botulinum neurotoxin A. *J. Am. Chem. Soc.* 139, 218–230. doi:10.1021/jacs.6b09534” were added to the manuscript to illustrate the point. Please see line 90 of the new manuscript.

5. Line 90: Critical literature is missing. Please either cite all original literature including crystal structure of toxin receptor complexes, or cite a recent comprehensive review focused on protein receptor binding.

The references [10] “Montecucco, C. (1986). How do tetanus and botulinum toxins bind to neuronal membranes?. *Trends Biochem. Sci.* 11, 314–317. doi:10.1016/0968-0004(86)90282-3”; [14] Dong, M. & Stenmark, P. The Structure and Classification of Botulinum Toxins. *Handb Exp Pharmacol* 263, 11–33 (2021). [15] Jin, R., Rummel, A., Binz, T. & Brunger, A. T. Botulinum neurotoxin B recognizes its protein receptor with high affinity and specificity. *Nature* 444, 1092–1095 (2006). [16] Chai, Q. et al. Structural basis of cell surface receptor recognition by botulinum neurotoxin B. *Nature* 444, 1096–1100 (2006). [17] Yao, G. et al. N-linked glycosylation of SV2 is required for binding and uptake of botulinum neurotoxin A. *Nat Struct Mol Biol* 23, 656–662 (2016). [18] Gustafsson, R., Zhang, S., Masuyer, G., Dong, M. & Stenmark, P. Crystal Structure of Botulinum Neurotoxin A2 in Complex with the Human Protein Receptor SV2C Reveals Plasticity in Receptor Binding. *Toxins (Basel)* 10, 153 (2018). [19] Benoit, R. M. et al. Structural basis for recognition of synaptic vesicle protein 2C by botulinum neurotoxin A. *Nature* 505, 108–111 (2014). [20] Benoit, R. M. et al. Crystal structure of the BoNT/A2 receptor-binding domain in complex with the luminal domain of its neuronal receptor SV2C. *Sci Rep* 7, 43588 (2017). have been added in the manuscript. Please see line 92 new manuscript.

6. Lines 112-116: this text and citation is misleading, as this test is not used for potency testing of pharmaceutical BoNTs. Even though BioSentinal has now developed an assay based on this technology, it is using a different cell line and possibly a different substrate

We agree that the paragraph was misleading so it was removed.

7. Line 117-120: This has been reviewed and discussed extensively. Please cite reference: Pellett S.Curr Top Microbiol Immunol. 2013;364:257-85.

The reference "Pellett S.Curr Top Microbiol Immunol 2013;364:257-85" was added. Please see line 118; ref 30.

8. Line 123: this work does not describe development of a new assay, but rather improvement of a current assay with a new endpoint using a reporter, which has previously been suggested (see ref 15 and Pellett S.Curr Top Microbiol Immunol. 2013;364:257-85.)

The text was corrected and the term "new" was removed from our sentence. Please see line 119 of the new manuscript.

9. Lines 125-126: Please describe whether other functional enzyme assays, and specifically cell based assays, have been developed using the split-Nanoluc luciferase technology

The references [33] Shetty, S. K., Walzem, R. L. & Davies, B. S. J. A novel NanoBiT-based assay monitors the interaction between lipoprotein lipase and GPIHBP1 in real time. *J Lipid Res* 61, 546–559 (2020). [34].Nguyen, L. P. *et al.* Establishment of a NanoBiT-Based Cytosolic Ca²⁺ Sensor by Optimizing Calmodulin-Binding Motif and Protein Expression Levels. *Mol Cells* 43, 909–920 (2020). [35].Li, B. *et al.* High-Throughput NanoBiT-Based Screening for Inhibitors of HIV-1 Vpu and Host BST-2 Protein Interaction. *Int J Mol Sci* 22, 9308 (2021). [36] Reyes-Alcaraz, A. *et al.* A NanoBiT assay to monitor membrane proteins trafficking for drug discovery and drug development. *Commun Biol* 5, 212 (2022). were added to the manuscript to illustrate this point. Please see lines 123-124 of the new manuscript.

10. Lines 130-135: This sentence should cite previous publications on assays using the exact same hiPSC derived neurons as are used in this study, and describing the same advantages just with a different endpoint. The use of hiPSC derived motoneurons is not new, and great sensitivity has been demonstrated previously

We agree that the use of hiPSC derived MNs and their sensitivity to BoNT has already been reported by several labs. Studies describing the use of these iPSC derived MN for BoNT testing have been acknowledged and cited in the introduction, please see line 118.

11. Lines 139-149: Please be more specific on the design of constructs. Please provide source sequence accessions used, primer sequences, more detail on the internal control. etc. It is completely unclear what BoNT sensor 4 is from this description alone – please refer to suppl. Fig.

A more detailed description of these constructs and their sequences has been added in the material and method section. Please see line 137 to 186 of the new manuscript.

12. Lines 151-154: Please indicate the exact viruses used

Supplementary information about the virus were added in the section lentiviral vector production. Please see line 189 to 192 of the new manuscript.

13. Line 167-168: Please provide more detail on dialysis – buffer volumes, number of changes, duration, temperature

Information was added in the section His tag sensor proteins purification. Please see lines 206-210.

14. Line 177: Please describe the negative controls

Information was added. Please see line 235 to 240 of the new manuscript.

15. Lines 181-185: The section title of rodent detection implies in vivo assays. In addition, much more detail is required here. How were cells prepared, from which animal species, how were the cells cultured, how were the cells exposed to lentivirus, etc.

The title was changed to “rodent cell assay” and we added information in this section. Please see line 243 to 251 of the new manuscript.

16. Line 218-219: This last sentence seems unnecessary

The sentence has been deleted.

17. Line 222-223: Please provide more information on the purchased toxins. Were they complex or 150 kDa purified BoNT? What was the specific activity? Was activity confirmed upon receipt? How were they stores?

We added information in method section called BoNTs. Please see line 213 to 226 of the new manuscript.

18. Line 227: What do the authors mean by the statement ‘All BoNTs were activated to more than 91% final purity’? How were the toxins activated?

Details on the procedure and the Reference [41] “Elliott et al., Sci. Adv. 2019; 5 : eaau7196 has been added in the Material and method section. This reference provides detailed information’s on the procedures for recombinant BoNT production and analysis.

19. Line 257: Please provide the source of GT1b

The source was provided. Please see line 313 of the new manuscript.

20. Lines 276-277: Incomplete sentence

The sentence was corrected. Please see line 304 to 305 of the new manuscript.

21. Lines 322-326 and Fig 1A: The construct of the split luciferase is unclear. Based on the text and figure legend, it would almost appear that the authors describe the SNARE target to be between the N and C terminal fragments of luciferase. Even though the figure shows the SNARE target is outside the luciferase gene, the circular design is confusing. Would it be possible to re-design the figure to show the fusion protein in a more descriptive manner, not as a circle?

Please also change the text to better describe the basis of the split luciferase assay. Why is the linker to the N-terminal luciferase longer, and so much longer than the SNARE target itself? The entire sequence of the construct should be provided in some manner, or deposited

Figure 1A was redesigned, and the entire sequences of the constructs are now provided in the Method section. Please see lines 137 to 186 of the new manuscript.

22. Line 326-327: ‘The remaining luciferase activity can be inversely 327 correlated with the toxin activity.’ At this point

in the ms, this should be phrased as a hypothesis

The sentence was corrected. Please see line 365 to 366 of the new manuscript.

23. Line 328: Please replace several with the number of constructs designed

The text was corrected. Please see line 366 of the new manuscript.

24. Line 331: Please explain why SV2C was added, since it is a receptor for the HC not the LC of BoNT We deleted SV2C data in our paper since it is not relevant to our cell-based assay.

25. Line 344: please add 'in an in vitro assay'
The sentence was suppressed.

26. Lines 335-346: It should be clearly stated that this section describes an in vitro assay using reduced BoNTs
We specified that at this stage of the study, we tested this sensor in an acellular assay. Please see line 365 to 366 of the new manuscript.

27. Figure 1c: the labels in the figure are quite small and difficult to see. Please increase the size of the labels. The figure legend states the assay was performed in duplicate, but one curve seems to show standard deviations. Please either show both curves for the duplicate assays or one representative curve if they are overlapping. In addition, please specify what duplicates refers to, were two independent assays conducted, two independent toxin dilution series, or the same dilution series in the same assay done in duplicate on the plate
For all the figures we have chosen the type face and optimum font size suggested by the journal: Arial, size 8pt . The figure legend was not precise. Indeed the experiment was done twice in duplicate, that's why we can calculate standard deviation.

28. Figure 2: please increase font
For all the figures we have chosen the type face and optimum font size suggested by the journal: Arial, size 8pt

29. Lines 363-365: As a common challenge of transfecting neurons with lentiviral vectors is cell death, please add data demonstrating cell viability and percentage of transfected cells.
The data showing cell death (Supplementary figure 2D) and transfection rate (Supplementary figure 2E-F) were added.

30. Figure 2B: It is unclear what is shown in the graph. The Y-axis is labeled as percent SNAP-25 cleavage, but based on the description this was not directly measured
On the first version of the paper the legend of the Y axis of figure 2B and 2C were unfortunately switched. The legend of the figure 2B was corrected. Figure 2B shows the decrease of the ratio percentage when we increase the amount of toxin.

31. Figure 2C: Similarly, the Y axis in this figure is labeled as ratio percentage, and while I can figure out what this means by flipping back and forth through the manuscript sections, it should be obvious from the figure alone. Please clarify labels and figure legend.
On the first version of the paper the legend of the Y axis of figure 2B and 2C were unfortunately switched. The legend of the figure 2C was corrected. Figure 2C shows increasing percentage of SNAP25 cleavage when we increase the amount of toxin.

32. Line 371: Ref 26 is just an abstract, not a peer reviewed publication. The first peer reviewed publication showing sensitivity of hiPSC derived motor neurons to BoNTs is ref 17, which should be cited here. Another reference that should be cited is Analysis of Motor Neurons Differentiated from Human Induced Pluripotent Stem Cells for the Use in Cell-Based Botulinum Neurotoxin Activity Assays. Schenke M, Schjeide BM, Püschel GP, Seeger B. Toxins (Basel). 2020 Apr 25;12(5):276. doi: 10.3390/toxins12050276.

Reference 26 of the first version of the paper was removed, and references suggested showing sensitivity of hiPSC derived motor neurons to BoNTs were added in the introduction. Please see line 118. At this stage of the manuscript (lines 405-408), we added reference 29 because we focused on the commercial model iCell Motor neurons..

33. Line 374: 'the majority' is a vague term. Please be more specific

The sentence was corrected. Please see line 410 to 411 of the new manuscript.

34. Figure 2D: Why does GFP stain in a punctate pattern similar to DAPI or ISLET1? Wouldn't the reporter be expected to be cytosolic or membrane bound?

To minimize the background signal, we decided to decrease the MOI of the transduction. At the time of exposure chosen for this picture, it was not possible to distinguish the staining in the neurites as you do at higher infection rate (supplementary figure 2B). To illustrate MN identity of our cells and infection rate we chose to reperform some experiments, so the picture we discussed was removed.

35. Lines 406-407: Why was transduction done at 48 h after plating, and the assay 6 days later?

We chose to transduce MN 48 h after plating to minimize cell death. The assay was performed 7 days after to let time to the cell to mature properly before adding the toxin.

36. Please define pEC50

pEC50 is defined in Mat and meth section in the luciferase assay section. Please see line 330-331 of the new manuscript.

37. Lines 414-415: Please also provide values showing the EC50 in Units of toxin activity, since this is a functional assay that measures toxin activity. Without this, the data cannot be compared to other literature

The value of EC50 was translated in Unit. Please see line 534 of the new manuscript.

38. Figure 3: An assay showing endogenous SNAP-25 cleavage by Western blot or ICC should be included for direct comparison, so that the sensitivity of this new endpoint can be evaluated

An assay showing endogenous SNAP-25 cleavage by Western blot is shown in figure 2C with an pEC 50 of 11,89 for BoNT/A (pEC 50= 12,57 with the BoNT sensor4).

39. Lines 426-427: 'increased' is vague. Please be more specific.

The sentence was rewritten. Please see line 472 to 474 of the new manuscript.

40. Figure 4: please increase label size

For all the figures we have chosen the type face and optimum font size suggested by the journal: Arial, size 8pt.

41. Line 424, ref 14: This was done using a different cell type

This was done on SiMA, SH-SY5Y, PC12, Neuro2A, NG108-15 cell lines but also on iCell neurons (from CDI). And authors showed that addition of Gt1b increase toxin sensitivity of these cells.

42. Line 427: Without quantifying the extracellular GT1b with and without external addition, these data do not show mechanisms, especially with only a 10% difference in SNAP-25 cleavage at only one toxin concentration

We have performed staining experiments showing that addition of GT1b to the media increase the amount of gangliosides at the surface of neurons. These new data are described in

the manuscript. Please see line 470 to 472 of the new manuscript. These data are shown in Supplementary Figure 6.

43. Lines 435-437: These conclusions are not supported by the two assays shown here. However, all steps of intoxication are required in a cell based assay by definition
The text was corrected. Please see line 482 to 484 of the new manuscript.

44. Line 445: Please change 'new assay' to 'a new endpoint for an existing assay' or 'an improvement on the hiPSC cell based assay'
The text was corrected. Please see line 489 of the new manuscript.

45. Line 450: please add citations for peer reviewed publications previously describing use of the exact same cells used here for the same purpose
Citations were added. Please see line 492 of the new manuscript.

46. Line 451: delete 'new'
The word has been removed.

47. Line 454, ref 17: the ref is appropriate but please see earlier comments on additional ref
Citations were added. Please see line 496 of the new manuscript.

48. Line 461: "moreover" seems misplaced here. The luminescent read-out for this specific assay is the only novel aspect of this study.
"Moreover" was removed.

49. Line 463: Why do the authors focus only on this ELISA read-out? There's other read-outs described including ICC, Western blot, other ELISA citations, neurotransmitter release...
We focused here on ELISA readout because it is the first and major readout used for an CB assay. Other read outs (fluorescent sensor, western blot or neurotransmitter release) are discussed in the following paragraph. Please see lines 510 to 530.

50. Lines 473-478: This reads more like a company defending its product than scientific analysis of benefits and limitations. Please re-write to highlight the unique aspects of each read-out, i.e. serotype specificity, versatility, specificity of endpoint, etc
The paragraph was rewritten and discussion about the sensitivity of our assay was discussed. Please see lines 526 to 541.

51. Line 491: This conclusion is too broad. The authors specifically show greater sensitivity of one human motor neurons cell model to BoNT/E. This does not mean that human neurons in general are more sensitive to BoNT/E than to BoNT/A. – please see ref 17
The paragraph was rewritten. In this part, we simply state that our results are consistent with previous data published by our group and these results matched with the first clinical trial in human made with BoNT/E. Please see lines 542 to 549.

52. Line 493-506: This is a long discussion comparing results from one citation that says 'data not shown' and states no difference was noted, to results from this study which used only one toxin concentration and noted a modest difference only. Unless these data are repeated with a dose response curve, conclusions can only be limited
This part of the discussion was deleted.

Reviewers' comments:

Reviewer #1 (Remarks to the Author):

The authors adequately dealt with all points addressed. There are no further concerns.

Reviewer #2 (Remarks to the Author):

I have carefully read the authors responses to each of my comments. The answers are satisfying provided that all correction were implemented in the manuscript.

Reviewer #3 (Remarks to the Author):

The manuscript is improved, however, a few items remain that should be addressed:

lines 104 ff: Why do the authors not list the second major cell-based assay used for potency determination of pharmaceuticals BoNTs, which is the hiPSC derived assay used by Merz? After all, the entire manuscript revolves a possible improvement in the readout of this specific assay.

line 313-316: A description of the cell-bound GT1b quantification is missing.

line 331: It is completely unclear why the authors are using this pEC50 value rather than an EC50 value. In addition, for all pEC50 values, Units should be indicated, presumably M based on the figures.

lines 531-539: Based on the authors statement, the assay is less sensitive than the hiPSC derived motoneurons sensitivity previously published. This should be discussed.

Fig 4: I still fail to understand the significance of the GT1b addition experiment. The result in cleavage is very small, and the quantification of the incorporated GT1b is still ambiguous as it is not well described (what total cell area was used? - Cell density seems greater in the picture to the right).

REVIEWERS' COMMENTS:

Reviewer #3 (Remarks to the Author):

The authors addressed all questions and comments appropriately.

Point by Point responses to reviewers

Manuscript number: COMMSBIO-21-1802

Manuscript title: Split luciferase-based assay to detect botulinum neurotoxins using human motor neurons derived from induced pluripotent stem cell.

Reviewer #3 (Remarks to the Author):

The manuscript is improved, however, a few items remain that should be addressed:

lines 104 ff: Why do the authors not list the second major cell-based assay used for potency determination of pharmaceuticals BoNTs, which is the hiPSC derived assay used by Merz? After all, the entire manuscript revolves a possible improvement in the readout of this specific assay.

The reference Eisele K-H & Mander G (2015) Gangliosides for standardizing and increasing the sensitivity of cells to botulinum neurotoxins in in vitro test systems.WO2015/124618 A1 was added to the references. Please see line 105.

line 313-316: A description of the cell-bound GT1b quantification is missing.

The description of the cell-bound GT1b quantification was added in the legend of the figure where these data were presented (i.e. supplementary figure 6).

line 331: It is completely unclear why the authors are using this pEC50 value rather than an EC50 value. In addition, for all pEC50 values, Units should be indicated, presumably M based on the figures.

We agree it is useful to indicate the EC50 values. These values have been added on each experiment used to determine the toxin potency, EC50 values were expressed in Molar (i.e. on figures 2C and 3B) and also converted in Unit, please see lines 424 and 460 to 461.

lines 531-539: Based on the authors statement, the assay is less sensitive than the hiPSC derived motor neurons sensitivity previously published. This should be discussed.

Discussion has been added. See line 534 to 546.

Fig 4: I still fail to understand the significance of the GT1b addition experiment. The result in cleavage is very small, and the quantification of the incorporated GT1b is still ambiguous as it is not well described (what total cell area was used? - Cell density seems greater in the picture to the right).

The difference in cleavage after GT1b addition while being small is measurable and this experiment demonstrates that the CB assay described here can be used as a tool to explore BoNT intoxication mechanisms.

The description of the GT1b incorporation quantification has been detailed in sup Fig 6 for clarification. Briefly, the mask staining was used to delimit the cell area and GT1b signal was calculated and reported per cell area.